# Minimax Multi-Target Conformal Prediction with Applications to Imaging Inverse Problems

**Jeffrey Wen**                                        *wen.254@buckeyemail.osu.edu*
*Department of Electrical and Computer Engineering*
*The Ohio State University*

**Rizwan Ahmad**                                       *rizwan.ahmad@osumc.edu*
*Department of Biomedical Engineering*
*The Ohio State University*

**Philip Schniter**                                    *schniter.1@osu.edu*
*Department of Electrical and Computer Engineering*
*The Ohio State University*

**Reviewed on OpenReview:** *https://openreview.net/forum?id=53FEYwDQK0*

## Abstract

In ill-posed imaging inverse problems, uncertainty quantification remains a fundamental challenge, especially in safety-critical applications. Recently, conformal prediction has been used to quantify the uncertainty that the inverse problem contributes to downstream tasks like image classification, image quality assessment, fat mass quantification, etc. While existing works handle only a scalar estimation target, practical applications often involve multiple targets. In response, we propose an asymptotically minimax approach to multi-target conformal prediction that provides tight prediction intervals while ensuring joint marginal coverage. We then outline how our minimax approach can be applied to multi-metric blind image quality assessment, multi-task uncertainty quantification, and multi-round measurement acquisition. Finally, we numerically demonstrate the benefits of our minimax method, relative to existing multi-target conformal prediction methods, using both synthetic and magnetic resonance imaging (MRI) data.

## 1 Introduction

Imaging inverse problems (Bertero et al., 2021) span a wide array of tasks, such as denoising, inpainting, accelerated magnetic resonance imaging (MRI), limited-angle computed tomography, phase retrieval, and image-to-image translation. In such problems, the objective is to recover a true image $x_0$ from noisy, incomplete, or distorted measurements $y_0 = \mathcal{A}(x_0)$. These problems tend to be ill-posed, in that many distinct hypotheses of $x_0$ can explain the collected measurements $y_0$. When perfect recovery of $x_0$ is difficult or impossible, uncertainty quantification (UQ) is critical to safely using/interpreting a given reconstruction $\widehat{x}_0$, especially in high-stakes fields like science or medicine (Chu et al., 2020; Banerji et al., 2023).

The field of image recovery has evolved significantly over the decades, and most contemporary approaches are based on deep learning (DL) (Arridge et al., 2019). Quantitatively, recent DL-based methods outperform classical methods on average and, qualitatively, they produce reconstructions that are sharp and detailed (Ongie et al., 2020). When the inverse problem is highly ill-posed, classical methods tend to produce recoveries with recognizable visual artifacts, from which it is relatively easy to gauge uncertainty. For example, radiologists receive explicit training in this regard (Virmani et al., 2015). In contrast, DL-based methods can hallucinate, i.e., generate recoveries that are visually plausible but differ from the truth in clinically or scientifically important ways (Cohen et al., 2018; Belthangady & Royer, 2019; Hoffman et al., 2021; Muckley

et al., 2021; Bhadra et al., 2021; Gottschling et al., 2023; Tivnan et al., 2024). This underscores the need for rigorous UQ, e.g., methods that provide statistical guarantees on estimates of $x_0$ or of some function $\mu(x_0)$.

For example, a recent line of work (Wen et al., 2024; Cheung et al., 2024) quantifies the imaging-induced uncertainty on downstream tasks such as pathology classification or fat-mass quantification. Defining the target $z_0$ as the output of the task applied to the (unknown) true image, they use conformal prediction (Vovk et al., 2005; Angelopoulos & Bates, 2023) to construct prediction intervals $\mathcal{C}$ that are statistically guaranteed to contain the target. In a related line of work, Wen et al. (2025) provides statistical guarantees on the quality of the reconstructed image $\widehat{x}_0$ relative to the true image, where "quality" is defined according to an arbitrary full-reference image-quality (FRIQ) metric like peak signal-to-noise ratio (PSNR) or structural similarity index measure (SSIM) (Wang et al., 2004). Defining the target as the FRIQ of $\widehat{x}_0$ relative to the (unknown) true $x_0$, they use conformal prediction to construct a bound on FRIQ that is statistically guaranteed.

While the above methods rigorously quantify the downstream impact of reconstruction uncertainty, they handle only a scalar target. In practice, one may want to consider multiple targets. For example, one may seek to identify multiple pathologies from a single recovery or to judge the quality of that recovery according to multiple metrics. Although multi-target conformal prediction methods have been proposed, they suffer from either limited interpretability (Messoudi et al., 2022; Feldman et al., 2023)(Rosenberg et al., 2023; Thurin et al., 2025; Klein et al., 2025; Braun et al., 2025), a lack of guaranteed joint coverage (Messoudi et al., 2021; Teneggi et al., 2023; Park & Cho, 2025), or some combination of overly conservative prediction intervals and/or high computational complexity (Messoudi et al., 2020; Diquigiovanni et al., 2022; Sampson & Chan, 2024; Sun & Yu, 2024), as we explain in the sequel.

We thus propose a new approach to multi-target conformal prediction. For problems with $K \geq 1$ targets, our goal is to ensure a notion of fairness between targets. With prediction intervals $\mathcal{C}_k$ and scalar targets $Z_{0,k}$ for $k = 1, \dots, K$, our method aims to ensure that no one "single-target coverage" $\Pr\{Z_{0,k} \in \mathcal{C}_k\}$ is favored over another, while also ensuring that all prediction intervals simultaneously contain their corresponding targets with a user-specified probability of $1 - \alpha$. To do this, we minimize the maximum single-target coverage under the joint-coverage constraint $\Pr\{\cap_k Z_{0,k} \in \mathcal{C}_k\} \leq 1 - \alpha$. Since the single-target coverage increases with the interval size $|\mathcal{C}_k|$, our approach additionally aims to prevent any prediction set $\mathcal{C}_k$ from being unnecessarily large. Our contributions are as follows:

1. Using a minimax formulation, we propose a novel multi-target conformal prediction approach with finite-sample marginal joint-coverage guarantees and low computational complexity.

2. We prove that our method is minimax in the limit of infinite tuning and calibration data.

3. For inverse problems, we propose a multi-round measurement acquisition scheme with marginal coverage guarantees on the final round.

4. We numerically compare our proposed method to several existing multi-target conformal prediction methods on a synthetic-data problem and four accelerated-MRI problems.

## 2 Background

### 2.1 Single-target conformal prediction

Conformal prediction (Vovk et al., 2005; Angelopoulos & Bates, 2023) is a general framework that enables one to construct uncertainty intervals with certain statistical guarantees for any black-box predictor. Importantly, it does not require any distributional assumptions on the data other than exchangeability, which allows for adoption in a broad range of applications. In this section, we briefly review the basics of conformal prediction, and in particular the computationally-efficient version known as split conformal prediction (Papadopoulos et al., 2002; Lei et al., 2018).

Suppose that we have a black-box model $h : \mathcal{U} \to \mathbb{R}$ that predicts a target $z_0 \in \mathbb{R}$ from features $u_0 \in \mathcal{U}$. The prediction $\widehat{z}_0 = h(u_0)$ may or may not be close to the true target $z_0$, but one can use conformal prediction to compute a prediction interval $\mathcal{C}_\lambda(\widehat{z}_0) \subset \mathbb{R}$ that contains $z_0$ with high probability. To compute this interval,

conformal prediction uses a dataset $\{(u_i, z_i)\}_{i=1}^n$ of feature–target pairs distinct from those used to train $h(\cdot)$. This dataset is converted to a calibration set $d_{\mathsf{cal}} \triangleq \{(\widehat{z}_i, z_i)\}_{i=1}^n$ using $\widehat{z}_i = h(u_i)$, and $d_{\mathsf{cal}}$ is used to find a $\widehat{\lambda}(d_{\mathsf{cal}})$ satisfying the marginal coverage guarantee (Lei & Wasserman, 2014)

$$\Pr\left\{Z_0 \in \mathcal{C}_{\widehat{\lambda}(D_{\mathsf{cal}})}(\widehat{Z}_0)\right\} \geq 1 - \alpha, \tag{1}$$

where $\alpha$ is a user-chosen error rate. Here and in the sequel, we use capital letters to denote random variables and lower-case letters to denote their realizations. In words, (1) guarantees that the unknown target $Z_0$ falls within the interval $\mathcal{C}_{\widehat{\lambda}(D_{\mathsf{cal}})}(\widehat{Z}_0)$ with probability at least $1-\alpha$ when averaged over the randomness in the test data $(Z_0, \widehat{Z}_0)$ and calibration data $D_{\mathsf{cal}}$.

The process of computing $\widehat{\lambda}(d_{\mathsf{cal}})$ is known as calibration. To calibrate, one first defines a nonconformity score $s(\widehat{z}_i, z_i)$. The choice of the nonconformity score function is quite flexible; it requires only that the score is higher when there is a worse agreement between $z_i$ and $\widehat{z}_i$. Common approaches include the absolute residual, locally-weighted residual (Lei et al., 2018), and conformalized quantile regression methods (Romano et al., 2019). The nonconformity score $s_i = s(\widehat{z}_i, z_i)$ is then computed for each sample pair $(\widehat{z}_i, z_i)$ in the calibration set $d_{\mathsf{cal}}$, and $\widehat{\lambda}(d_{\mathsf{cal}})$ is chosen as

$$\widehat{\lambda}(d_{\mathsf{cal}}) \triangleq \mathrm{EmpQuant}\left(\frac{\lceil(1-\alpha)(n+1)\rceil}{n}; s_1, \ldots, s_n\right), \tag{2}$$

which is a slightly more conservative quantile than the $1-\alpha$ quantile. With $\widehat{\lambda}(d_{\mathsf{cal}})$ computed, the prediction interval for the $i$th sample is simply defined as

$$\mathcal{C}_{\widehat{\lambda}(d_{\mathsf{cal}})}(\widehat{z}_i) = \left\{z : s(\widehat{z}_i, z) \leq \widehat{\lambda}(d_{\mathsf{cal}})\right\}. \tag{3}$$

Following this design, the marginal coverage guarantee (1) holds when $(\widehat{Z}_0, Z_0), (\widehat{Z}_1, Z_1), \ldots, (\widehat{Z}_n, Z_n)$ are statistically exchangeable (Vovk et al., 2005), a weaker condition than i.i.d. Under the additional assumption that the nonconformity scores $S_0, S_1, \ldots, S_n$ are almost surely distinct, the coverage can also be upper bounded (Romano et al., 2019) by

$$\Pr\left\{Z_0 \in \mathcal{C}_{\widehat{\lambda}(D_{\mathsf{cal}})}(\widehat{Z}_0)\right\} \leq 1 - \alpha + \frac{1}{n+1}. \tag{4}$$

## 2.2 Application to imaging inverse problems

In imaging inverse problems, conformal prediction has emerged as a tool to quantify the uncertainty in image recovery. Several approaches (Angelopoulos et al., 2022b; Horwitz & Hoshen, 2022; Teneggi et al., 2023; Kutiel et al., 2023; Narnhofer et al., 2024) use conformal prediction to construct, for each individual pixel, an interval that is guaranteed to contain the true pixel value with high probability. For quantifying multi-pixel uncertainty, Belhasin et al. (2023) propose to compute conformal intervals on the principal components of the posterior covariance matrix, and Sankaranarayanan et al. (2022) construct conformal intervals for semantic attributes in the latent space of a disentangled generative adversarial network.

Although these notions of uncertainty are interesting to consider, they don't directly quantify the impact of recovery errors on downstream imaging tasks such as image classification, image quality assessment, and quantitative imaging. Consequently, task-based image uncertainty methods like (Wen et al., 2024; Cheung et al., 2024; Wen et al., 2025) have been proposed. We now briefly review these methods using a unified notational framework.

Both Wen et al. (2024) and Cheung et al. (2024) quantify the uncertainty in estimating $\mu(x_0) \in \mathbb{R}$ given the measurements $y_0$. In Wen et al. (2024) $\mu(\cdot)$ is a soft-output classifier, and in Cheung et al. (2024) it is a fat-mass quantifier, but in either case the target is set at $z_0 = \mu(x_0)$. Assuming access to an approximate posterior sampler $g(\cdot, \cdot)$ that generates $c$ samples $\{\widetilde{x}_i^{(j)}\}_{j=1}^c$ per measurement vector $y_i$ via $\widetilde{x}_i^{(j)} = g(y_i, \widetilde{v}_i^{(j)})$ using i.i.d code vectors $\widetilde{v}_i^{(j)} \sim \mathcal{N}(0, I)$, the prediction is computed as

$$\widehat{z}_i = [\widehat{z}_i^{(1)}, \ldots, \widehat{z}_i^{(c)}]^\top = [\mu(\widetilde{x}_i^{(1)}), \ldots, \mu(\widetilde{x}_i^{(c)})]^\top \in \mathbb{R}^c. \tag{5}$$

Several nonconformity scores can be used, but here we describe only the version with the conformalized quantile regression (CQR) method of Romano et al. (2019). With CQR, the $\frac{\alpha}{2}$ and $1 - \frac{\alpha}{2}$ empirical quantiles are computed as

$$\widehat{q}_{\frac{\alpha}{2}}(\widehat{z}_i) = \text{EmpQuant}\left(\frac{\alpha}{2}; \widehat{z}_i^{(1)}, \ldots, \widehat{z}_i^{(c)}\right) \quad \text{and} \quad \widehat{q}_{1-\frac{\alpha}{2}}(\widehat{z}_i) = \text{EmpQuant}\left(1 - \frac{\alpha}{2}; \widehat{z}_i^{(1)}, \ldots, \widehat{z}_i^{(c)}\right), \qquad (6)$$

respectively, and the nonconformity score is defined as

$$s(\widehat{z}_i, z_i) = \max\left\{\widehat{q}_{\frac{\alpha}{2}}(\widehat{z}_i) - z_i, z_i - \widehat{q}_{1-\frac{\alpha}{2}}(\widehat{z}_i)\right\}, \qquad (7)$$

after which the prediction interval $\mathcal{C}_{\widehat{\lambda}(d_{\text{cal}})}(\widehat{z}_i)$ is constructed as in (3). Because this prediction interval changes with $y_i$, it is said to be "adaptive" (Lei et al., 2018). In any case, it satisfies the marginal coverage guarantee in (1) when $(\widehat{Z}_0, Z_0), (\widehat{Z}_1, Z_1), \ldots, (\widehat{Z}_n, Z_n)$ are statistically exchangeable.

In related work, Wen et al. (2025) seek to estimate the FRIQ (e.g., PSNR, SSIM, etc.) $m(\widehat{x}_0, x_0)$ of an image recovery $\widehat{x}_0 = f(y_0)$ relative to the true image $x_0$ when given access to measurements $y_0$ but not $x_0$ itself. To do so, they set the target as $z_0 = m(\widehat{x}_0, x_0)$ and use an approximate posterior sampler that generates $c$ samples $\{\widetilde{x}_i^{(j)}\}_{j=1}^c$ per measurement vector $y_i$ to compute the prediction

$$\widehat{z}_i = [m(\widehat{x}_i, \widetilde{x}_i^{(1)}), \ldots, m(\widehat{x}_i, \widetilde{x}_i^{(c)})]^\top \in \mathbb{R}^c. \qquad (8)$$

Wen et al. (2025) then used empirical quantiles to construct a one-sided prediction interval $\mathcal{C}_\lambda(\cdot)$ to either lower- or upper-bound the FRIQ, as appropriate. Note that $m(\widehat{x}_i, \cdot)$ can be viewed as a recovery-conditioned task. To maintain consistency with other task-based approaches, when discussing FRIQ estimation in the sequel, we construct two-sided intervals using (3) with the nonconformity score from (7).

## 2.3 Multi-target conformal prediction

As discussed in Section 1, one may be interested in conformal prediction of several targets, which we combine into a multi-dimensional target vector $[z_{i,1}, \ldots, z_{i,K}] = z_i \in \mathbb{R}^K$. We focus on the case where one is given a prediction $\widehat{z}_0 \in \mathbb{R}^K$ of unknown test $z_0 \in \mathbb{R}^K$, along with a calibration set $d_{\text{cal}} = \{(\widehat{z}_i, z_i)\}_{i=1}^n$, and the goal is to compute $K$ prediction intervals $\{\mathcal{C}_{\widehat{\lambda}(D_{\text{cal}}),k}(\widehat{Z}_0)\}_{k=1}^K$ that satisfy the joint marginal coverage guarantee

$$\Pr\left\{\cap_{k=1}^K Z_{0,k} \in \mathcal{C}_{\widehat{\lambda}(D_{\text{cal}}),k}(\widehat{Z}_0)\right\} \geq 1 - \alpha. \qquad (9)$$

This guarantee ensures that all target components simultaneously lie within their respective prediction intervals with a probability at least $1-\alpha$ over the randomness in the calibration and test data.

Variations on (9) are possible, such as minimizing a risk that allows a fraction of the target components to lie outside their prediction intervals (Teneggi et al., 2023). Likewise, while (9) can be interpreted as constructing a hyper-rectangle in $\mathbb{R}^K$ that contains $Z_0$ with high probability, it is possible to construct non-rectangular regions, such as ellipsoidal regions (Messoudi et al., 2022) or more complicated regions defined by the outputs of a conditional variational auto-encoder (Feldman et al., 2023), vector quantile regressor (Rosenberg et al., 2023), optimal-transport map (Thurin et al., 2025; Klein et al., 2025), or a volume-minimizing mapping (Braun et al., 2025). Although these non-rectangular regions can give smaller uncertainty volumes, they are less interpretable for the applications we consider, since the uncertainty interval on one target component will depend on the values of the other target components.

Inspired by (9), several approaches have been proposed to construct prediction intervals. Messoudi et al. (2020) assume that the nonconformity score components are statistically independent, so that when the components are individually calibrated to yield an error-rate of $\alpha_1$, the joint error-rate $\alpha$ will equal $1 - (1-\alpha_1)^K$. This allows setting $\alpha_1 = 1 - (1-\alpha)^{1/K}$ to meet a desired joint error-rate of $\alpha$. However, the independence assumption may not hold in practice, where one could encounter $K$ dependent score components that yield a joint error-rate $> 1 - (1-\alpha_1)^K$, in which case the joint-coverage guarantee (9) would be violated. But even when the joint coverage holds, we show in Section 5 that the prediction intervals from Messoudi et al. (2020) are overly conservative.

Another line of work uses copulas (Nelsen, 2006) to model the statistical dependency between the score components $\{s_{i,k}\}_{k=1}^K$, where $s_{i,k} = s_k(\widehat{z}_{i,k}, z_{i,k})$. Given a random vector $S = [S_1, \ldots, S_K]$ with marginal CDFs $F_{S_k}(s_k) = \Pr\{S_k \leq s_k\}$, the copula of $S$ is defined as the function $C_S : [0,1]^K \to [0,1]$ for which

$$C_S(v_1, \ldots, v_K) = \Pr\{F_{S_1}(S_1) \leq v_1, \ldots, F_{S_K}(S_K) \leq v_K\}. \tag{10}$$

That is, the probability integral transform is used to convert each marginal $S_k$ into a uniform random variable $V_k = F_{S_k}(S_k)$ and the copula $C_S$ is the joint CDF of $V = [V_1, \ldots, V_K]$. Messoudi et al. (2021) approximate the copula as $\widehat{C}_S$ using the empirical marginal CDFs $\widehat{F}_{S_k}$ computed using the calibration data, and then search for a $\widehat{v} \in [0,1]$ for which $\widehat{C}_S(\widehat{v}, \ldots, \widehat{v}) \geq 1-\alpha$. The corresponding $\widehat{\lambda}(d_{\mathsf{cal}}) = [\widehat{F}_{S_1}^{-1}(\widehat{v}), \ldots, \widehat{F}_{S_K}^{-1}(\widehat{v})] \in \mathbb{R}^K$ then approximately satisfies $F_S(\widehat{\lambda}(d_{\mathsf{cal}})) \geq 1-\alpha$, and $\widehat{\lambda}_k(d_{\mathsf{cal}})$ can be used to define a prediction interval for $\widehat{z}_{i,k}$ via (3). However, they provide no coverage guarantees on these intervals. Sun & Yu (2024) and Park & Cho (2025) instead search for a $\widehat{v} \in [0,1]^K$ for which

$$\widehat{v} = \arg\min_{v \in [0,1]^K} \sum_{k=1}^K v_k \text{ such that } \widehat{C}_S(v) \geq 1 - \alpha, \tag{11}$$

after which they set $\widehat{\lambda}(d_{\mathsf{cal}}) = [\widehat{F}_{S_1}^{-1}(\widehat{v}_1), \ldots, \widehat{F}_{S_K}^{-1}(\widehat{v}_K)] \in \mathbb{R}^K$. Park & Cho (2025) use semiparametric vine copulas and provide coverage guarantees only in the limit of infinite calibration data. Sun & Yu (2024) use the empirical copula together with conformal marginal CDFs (Vovk et al., 2017) and obtain a finite-data coverage guarantee similar to (9). But (11) provides no incentive for balancing coverage across targets (as we will see in Section 5), and there's no clear way to solve the non-convex optimization in (11). Even approximately solving (11) is computationally expensive (e.g., Sun & Yu (2024) use gradient descent).

As an alternative, Diquigiovanni et al. (2022) and Sampson & Chan (2024) propose to combine the score components $\{s_{i,k}\}_{k=1}^K$ into a single score via

$$s_i = \max\{s_{i,1}, \ldots, s_{i,K}\}. \tag{12}$$

Using calibration $\{s_i\}_{i=1}^n$ with (2) and extending (3) to component-wise intervals

$$\mathcal{C}_{\widehat{\lambda}(d_{\mathsf{cal}}),k}(\widehat{z}_i) = \left\{ z : s_k(\widehat{z}_{i,k}, z) \leq \widehat{\lambda}(d_{\mathsf{cal}}) \right\}, \quad k = 1, \ldots, K, \tag{13}$$

the arguments from Vovk et al. (2005) imply that the joint-coverage guarantee (9) and upper bound (4) both hold under the usual exchangeability assumption. However, Sampson & Chan (2024) note that this approach can disproportionately favor the target components with larger nonconformity scores, causing the prediction intervals of the other components to be overly conservative. To mitigate this issue, they propose to scale the nonconformity scores to a common range. To do so, they first train a pair of quantile regressors $\widehat{q}_{\frac{\alpha}{2},k}(\cdot)$ and $\widehat{q}_{1-\frac{\alpha}{2},k}(\cdot)$ that estimate the $\frac{\alpha}{2}$ and $1-\frac{\alpha}{2}$ quantile of $Z_{i,k}$, respectively, for each $k \in \{1, \ldots, K\}$, and then form the scaled nonconformity scores

$$s_{i,k} = \max\left\{\widehat{q}_{\frac{\alpha}{2},k}(\widehat{z}_i) - z_{i,k}, z_{i,k} - \widehat{q}_{1-\frac{\alpha}{2},k}(\widehat{z}_i)\right\} \underbrace{\frac{\widehat{q}_{1-\frac{\alpha}{2},1}(\widehat{z}_i) - \widehat{q}_{\frac{\alpha}{2},1}(\widehat{z}_i)}{\widehat{q}_{1-\frac{\alpha}{2},k}(\widehat{z}_i) - \widehat{q}_{\frac{\alpha}{2},k}(\widehat{z}_i)}}_{\text{Scale relative to 1st target}}, \tag{14}$$

where $\widehat{z}_i = u_i$. This helps to balance the single-target coverages $\Pr\{Z_{0,k} \in \mathcal{C}_{\widehat{\lambda}(D_{\mathsf{cal}}),k}(\widehat{Z}_0)\}$ across $k \in \{1, \ldots, K\}$ while ensuring the joint-coverage guarantee (9) and upper bound (4). In Section 3, we show how to obtain a better balance.

In the broader scope of distribution-free UQ, the Learn-Then-Test (LTT) framework of Angelopoulos et al. (2022a) provide an alternative approach to handling multiple targets that uses multiple hypothesis testing. Although LTT is generally used for cases in which there are multiple notions of risk, our case involves only a single risk and thus LTT would provide a guarantee of the form

$$\Pr\left[\Pr\left\{ \cap_{k=1}^K Z_{0,k} \in \mathcal{C}_{\widehat{\lambda}(D_{\mathsf{cal}}),k}(\widehat{Z}_0) \,\middle|\, D_{\mathsf{cal}} \right\} \geq 1 - \alpha\right] \geq 1 - \delta, \tag{15}$$

where $\alpha, \delta \in (0, 1)$ are each user-selected error rates. In (15), the inner probability is over the randomness in the test data while the outer probability is over the randomness in the calibration data. Since the LTT guarantee takes a different form than (9), the LTT procedure is not directly comparable to any of the previously mentioned multi-target methods. Also, the use of two user-selectable error rates in (15) complicates the design. In this paper, we focus only on methods that provide joint-coverage guarantees of the form (9).

## 3 Minimax multi-target conformal prediction

In this section, we propose a new approach to multi-target conformal prediction using a minimax formulation. We first formulate the minimax problem using random variables. Then we present the finite-sample version of our approach, which manifests as an instance of split conformal prediction. Finally we prove that the finite-sample solution converges to the solution of the original minimax problem as the number of samples grows to infinity.

### 3.1 Random variable perspective

To build intuition, we first consider the design of prediction sets when the targets and predictions are modeled as random variables $Z = [Z_1, \ldots, Z_K] \in \mathbb{R}^K$ and $\widehat{Z} = [\widehat{Z}_1, \ldots, \widehat{Z}_K]$, respectively. For the $k$th component, suppose that the nonconformity score function is $s_k(\cdot, \cdot)$ and the prediction set is constructed as $\mathcal{C}_{\widehat{\zeta}_k}(\widehat{Z}_k) \triangleq \{z : s_k(\widehat{Z}_k, z) \leq \widehat{\zeta}_k\}$, where $\widehat{\zeta}_k$ is a design variable. Then the "single-target coverage" of the $k$th component will be

$$\Pr\left\{Z_k \in \mathcal{C}_{\widehat{\zeta}_k}(\widehat{Z}_k)\right\} = \Pr\left\{Z_k \in \{z : s_k(\widehat{Z}_k, z) \leq \widehat{\zeta}_k\}\right\} = \Pr\left\{s_k(\widehat{Z}_k, Z_k) \leq \widehat{\zeta}_k\right\}. \tag{16}$$

Using $S_k \triangleq s_k(\widehat{Z}_k, Z_k)$, we can write the single-target coverage more succinctly as

$$\Pr\left\{S_k \leq \widehat{\zeta}_k\right\} = F_{S_k}(\widehat{\zeta}_k), \tag{17}$$

where $F_{S_k}(\widehat{\zeta}_k)$ is the CDF of $S_k$ evaluated at $\widehat{\zeta}_k$. Similarly, the joint coverage of all $K$ components will be

$$\Pr\left\{\cap_{k=1}^K Z_k \in \mathcal{C}_{\widehat{\zeta}_k}(\widehat{Z}_k)\right\} = \Pr\left\{\cap_{k=1}^K S_k \leq \widehat{\zeta}_k\right\}. \tag{18}$$

For a given joint miscoverage rate of $\alpha$, we'd like to find a tuple $(\widehat{\zeta}_1, \ldots, \widehat{\zeta}_K)$ that ensures

$$\Pr\left\{\cap_{k=1}^K S_k \leq \widehat{\zeta}_k\right\} \geq 1 - \alpha. \tag{19}$$

But, in general, many $(\widehat{\zeta}_1, \ldots, \widehat{\zeta}_K)$ will yield the same value of $\Pr\{\cap_{k=1}^K S_k \leq \widehat{\zeta}_k\}$, and for some choices of $(\widehat{\zeta}_1, \ldots, \widehat{\zeta}_K)$, a portion of the prediction intervals $\mathcal{C}_{\widehat{\zeta}_k}(\widehat{Z}_k)$ can be overly conservative. Since a larger $\Pr\{S_k \leq \widehat{\zeta}_k\}$ generally corresponds to a larger prediction interval $\mathcal{C}_{\widehat{\zeta}_k}(\widehat{Z}_k)$ due to their monotonically non-decreasing relationship, we propose to design prediction intervals that minimize the maximum single-target coverage while ensuring joint coverage, i.e.,

$$(\widehat{\zeta}_1, \ldots, \widehat{\zeta}_K) = \arg\min_{\zeta_1, \ldots, \zeta_K} \max_k \Pr\{S_k \leq \zeta_k\} \text{ s.t. } \Pr\{\cap_{k=1}^K S_k \leq \zeta_k\} \geq 1 - \alpha. \tag{20}$$

Using (17) and the fact that the CDF is non-decreasing, we can restate (20) as

$$(\widehat{\zeta}_1, \ldots, \widehat{\zeta}_K) = \arg\min_{\zeta_1, \ldots, \zeta_K} \max_k F_{S_k}(\zeta_k) \text{ s.t. } \Pr\{\cap_{k=1}^K F_{S_k}(S_k) \leq F_{S_k}(\zeta_k)\} \geq 1 - \alpha, \tag{21}$$

and further restate it using $\lambda_k \triangleq F_{S_k}(\zeta_k)$ as

$$(\widehat{\lambda}_1, \ldots, \widehat{\lambda}_K) = \arg\min_{\lambda_1, \ldots, \lambda_K} \max_k \lambda_k \text{ s.t. } \Pr\{\cap_{k=1}^K F_{S_k}(S_k) \leq \lambda_k\} \geq 1 - \alpha. \tag{22}$$

Although the solution to (22) may not be unique, it suffices to find a single minimax $(\widehat{\lambda}_1, \ldots, \widehat{\lambda}_K)$. Towards this aim, observe that $\Pr\{\cap_{k=1}^K F_{S_k}(S_k) \leq \lambda_k\}$ is monotonically non-decreasing with respect to any $\lambda_k$. Thus, given any tuple $(\lambda_1, \ldots, \lambda_K)$ that satisfies the constraint in (22), the tuple $(\lambda', \ldots, \lambda')$ for $\lambda' \triangleq \max_k \lambda_k$ also satisfies the constraint while simultaneously yielding the same value of the objective "$\max_k \lambda_k$." This implies that, without loss of minimax optimality, we can reframe (22) as a search for a single parameter $\widehat{\lambda}$:

$$\widehat{\lambda} = \arg\min_\lambda \lambda \ \text{ s.t. } \ \Pr\{\cap_{k=1}^K F_{S_k}(S_k) \leq \lambda\} \geq 1 - \alpha. \tag{23}$$

Although (22) bears some similarities to the copula methodology in (11), note that (22) uses a max where (11) uses a sum. Also, (22) can be reduced to a simple one-dimensional search (23), whereas the non-convex (11) involves an expensive multi-variable optimization.

## 3.2 Finite-sample case

We now adapt (23) to the practical case where the $S_k$ distributions are unknown and must be learned. For this purpose, we assume access to "tuning" data $\{(u_i, z_i)\}_{i=n+1}^{n+n_{\text{tune}}}$ that is distinct from the data $\{(u_i, z_i)\}_{i=1}^n$ used for split conformal prediction and from the data used to train the predictor that generates $\widehat{z}_i$.

We propose to do the following for each target component $k \in \{1, \ldots, K\}$. First, we construct the set $d_{\text{tune},k} \triangleq \{(\widehat{z}_{i,k}, z_{i,k})\}_{i=n+1}^{n+n_{\text{tune}}}$ and compute the nonconformity score $s_{i,k}$ for all samples $i$ in $d_{\text{tune},k}$. Using these nonconformity scores, we compute the empirical CDF $\widehat{F}_{S_k}(\cdot)$, where

$$\widehat{F}_{S_k}(\zeta) = \frac{|\{s_{i,k} : s_{i,k} \leq \zeta, \ i = n+1, \ldots, n+n_{\text{tune}}\}|}{n_{\text{tune}}}. \tag{24}$$

Next, we compute the nonconformity scores $s_{i,k}$ for all samples $i$ in the *calibration* set $d_{\text{cal},k} \triangleq \{(\widehat{z}_{i,k}, z_{i,k})\}_{i=1}^n$ and apply the learned $\widehat{F}_{S_k}(\cdot)$ to obtain the transformed calibration scores

$$\overline{s}_{i,k} \triangleq \widehat{F}_{S_k}(s_{i,k}) \text{ for } i = 1, \ldots, n. \tag{25}$$

Because $\overline{s}_{i,k} \in [0, 1]$, the scores $\{\overline{s}_{i,k}\}_{i=1}^n$ are implicitly normalized across $k \in \{1, \ldots, K\}$. Finally, we take the maximum across components,

$$\overline{s}_i \triangleq \max(\overline{s}_{i,1}, \ldots, \overline{s}_{i,K}), \tag{26}$$

and from these $\{\overline{s}_i\}_{i=1}^n$ compute $\widehat{\lambda}(d_{\text{cal}})$ in the same manner as (2):

$$\widehat{\lambda}(d_{\text{cal}}) = \text{EmpQuant}\left(\frac{\lceil(1-\alpha)(n+1)\rceil}{n}; \overline{s}_1, \ldots, \overline{s}_n\right). \tag{27}$$

The target-domain prediction intervals can then be constructed as

$$\mathcal{C}_{\widehat{\lambda}(d_{\text{cal}}),k}(\widehat{z}_i) = \{z : \widehat{F}_{S_k}(s_k(\widehat{z}_{i,k}, z)) \leq \widehat{\lambda}(d_{\text{cal}})\}, \quad k = 1, \ldots, K. \tag{28}$$

Since the tuning data used to construct $\widehat{F}_{S_k}(\cdot)$ is distinct from the calibration data, our method follows the framework of split conformal prediction. By taking the max of the transformed scores, we ensure the inclusion of all the scores, and thus enjoy the finite-sample joint marginal coverage guarantee of (9), as summarized in the following lemma:

**Lemma 1.** *For any $\alpha \in (0, 1)$, the prediction intervals $\{\mathcal{C}_{\widehat{\lambda}(D_{\text{cal}}),k}(\widehat{Z}_0)\}_{k=1}^K$ from (28) obey*

$$\Pr\left\{\cap_{k=1}^K Z_{0,k} \in \mathcal{C}_{\widehat{\lambda}(D_{\text{cal}}),k}(\widehat{Z}_0)\right\} \geq 1 - \alpha \tag{29}$$

*when $(\widehat{Z}_0, Z_0), (\widehat{Z}_1, Z_1), \ldots, (\widehat{Z}_n, Z_n)$ are statistically exchangeable.*

The guarantee in Lemma 1 is similar to that in Sampson & Chan (2024) and stronger than that in Sun & Yu (2024), since the latter requires exchangeability of the test, calibration, *and tuning* samples. (In fact the

---

**Algorithm 1** Minimax-based conformal prediction of test target $z_0 \in \mathbb{R}^K$ from feature vector $u_0 \in \mathcal{U}$.

---

**Require:** Error rate $\alpha \in (0, 1)$.
             Test feature vector $u_0$.
             Prediction model $h : \mathcal{U} \to \mathbb{R}^K$.
             Data $\{(u_i, z_i)\}_{i=1}^{n+n_\mathsf{tune}}$ not used to train $h(\cdot)$.
             Nonconformity score functions $s_k : \mathbb{R} \times \mathbb{R} \to \mathbb{R}$ for $k = 1, \ldots, K$.
1: Compute the predictions $\{\widehat{z}_i\}_{i=0}^{n+n_\mathsf{tune}}$ using $\widehat{z}_i = h(u_i)$.
2: **for** $k = 1, \ldots, K$ **do**
3:     Compute the nonconformity scores $\{s_{i,k}\}_{i=1}^{n+n_\mathsf{tune}}$ using $s_{i,k} = s_k(\widehat{z}_{i,k}, z_{i,k})$.
4:     Compute the empirical CDF $\widehat{F}_{S_k}(\cdot)$ using $\widehat{F}_{S_k}(\zeta) = \dfrac{|\{s_{i,k} : s_{i,k} \leq \zeta,\ i = n+1, \ldots, n+n_\mathsf{tune}\}|}{n_\mathsf{tune}}$.
5:     Compute the transformed nonconformity scores $\{\overline{s}_{i,k}\}_{i=1}^{n}$ using $\overline{s}_{i,k} = \widehat{F}_{S_k}(s_{i,k})$.
6: Compute the component-maximized scores $\{\overline{s}_i\}_{i=1}^{n}$ using $\overline{s}_i = \max(\overline{s}_{i,1}, \ldots, \overline{s}_{i,K})$.
7: Compute the threshold $\widehat{\lambda}(d_\mathsf{cal}) = \mathrm{EmpQuant}\left(\dfrac{\lceil (1-\alpha)(n+1) \rceil}{n}; \overline{s}_1, \ldots, \overline{s}_n\right)$.
8: Compute the prediction intervals $\mathcal{C}_{\widehat{\lambda}(d_\mathsf{cal}),k}(\widehat{z}_i) = \left\{z : \widehat{F}_{S_k}\big(s_k(\widehat{z}_{i,k}, z)\big) \leq \widehat{\lambda}(d_\mathsf{cal})\right\}$ for $k = 1, \ldots, K$.
9: **return** Prediction interval $\mathcal{C}_{\widehat{\lambda}(d_\mathsf{cal}),k}(\widehat{z}_0)$ on unknown $z_{0,k}$ for each $k = 1, \ldots, K$.

---

proof in Sun & Yu (2024) makes the stronger assumption that the test, calibration, and tuning samples are i.i.d.) In other words, the finite-sample guarantee in Lemma 1 is conditional on the tuning samples whereas that in Sun & Yu (2024) is not. Furthermore, we demonstrate in Section 5 that, as a result of our minimax formulation, our coverages tend to be more balanced (across targets) than those of Sampson & Chan (2024) and Sun & Yu (2024), which prevents our prediction intervals from being unnecessarily large.

Algorithm 1 summarizes our proposed split-conformal prediction procedure.

### 3.3 Asymptotically minimax

Because the proposed finite-sample methodology uses the empirical CDF (24) in place of the true CDF $F_{S_k}$ and the empirical quantile (26)-(27) in place of the true quantile in (23), one may wonder how the calibration parameter $\widehat{\lambda}(d_\mathsf{cal})$ in (27) relates to the minimax $\widehat{\lambda}$ in (23). Below we show that $\widehat{\lambda}(d_\mathsf{cal})$ converges to $\widehat{\lambda}$ in the limit of infinite tuning and calibration data.

**Theorem 2.** *For each target component $k = 1, \ldots, K$, suppose that the nonconformity scores $\{S_{i,k}\}_{i=1}^{n+n_\mathsf{tune}}$ are i.i.d with CDF $F_{S_k}(\cdot)$ and, for $T \triangleq \max_k F_{S_k}(S_k)$, suppose that $F_T(\cdot)$ is continuous and strictly increasing at the $(1-\alpha)$-level quantile of $T$. Then $\widehat{\lambda}(d_\mathsf{cal})$ from (27) converges to $\widehat{\lambda}$ from (23) almost surely as $n \to \infty$ and $n_\mathsf{tune} \to \infty$.*

See Appendix A for a proof.

## 4 Applications of multi-target conformal prediction in imaging

In Section 2.2, we described several applications of conformal prediction to single-target UQ in imaging inverse problems. In this section, we propose several applications of conformal prediction to multi-target UQ in imaging inverse problems.

We now establish a common notation that can be used across several applications. Consider a target vector $z_i \in \mathbb{R}^K$, prediction matrix $\widehat{z}_i \in \mathbb{R}^{K \times c}$, and nonconformity score $s_{i,k} = s_k(\widehat{z}_{i,k}, z_{i,k})$ for the $k$th target and $i$th sample. With CQR, the nonconformity score would be computed as

$$s_{i,k} = s_k(\widehat{z}_{i,k}, z_{i,k}) = \max\left\{\widehat{q}_{\frac{\alpha}{2},k}(\widehat{z}_{i,k}) - z_{i,k}, z_{i,k} - \widehat{q}_{1-\frac{\alpha}{2},k}(\widehat{z}_{i,k})\right\} \tag{30}$$

with

$$\widehat{q}_{\frac{\alpha}{2},k}(\widehat{z}_{i,k}) = \mathrm{EmpQuant}\left(\frac{\alpha}{2}; \widehat{z}_{i,k}^{(1)}, \ldots, \widehat{z}_{i,k}^{(c)}\right) \quad \text{and} \quad \widehat{q}_{1-\frac{\alpha}{2},k}(\widehat{z}_{i,k}) = \mathrm{EmpQuant}\left(1 - \frac{\alpha}{2}; \widehat{z}_{i,k}^{(1)}, \ldots, \widehat{z}_{i,k}^{(c)}\right). \tag{31}$$

With this problem setup, the proposed minimax method from Section 3, or any of the existing multi-target approaches discussed in Section 2.3, can be applied to generate prediction sets $\mathcal{C}_{\widehat{\lambda}(d_{\mathsf{cal}}),k}(\widehat{z}_0)$ for target indices $k = 1, \ldots, K$. When the joint-coverage guarantee in (9) holds, all $K$ prediction sets will simultaneously include the corresponding true-target values with a probability of at least $1-\alpha$. We emphasize that this guarantee holds regardless of the quality of the approximate posterior sampler used to generate $\{\widehat{z}_{i,k}\}_{j=1}^c$. As this sampler gets worse, the prediction sets will get looser but the guarantee will still hold. Below, we describe how to construct these quantities in different applications.

## 4.1 Multi-metric blind FRIQ assessment

We first consider blind FRIQ assessment, where the goal is to estimate the FRIQ of a reconstruction $\widehat{x}_0 = f(y_0)$ relative to the true image $x_0$, given measurements $y_0 = \mathcal{A}(x_0)$ but no direct access to $x_0$. Whereas Section 2.2 discussed the use of a single FRIQ metric, one may instead be interested in assessing image quality according several FRIQ metrics, since different metrics may be complementary (Wang, 2011).

Consider the case of $K$ FRIQ metrics $\{m_k(\cdot, \cdot)\}_{k=1}^K$. To extend the conformal prediction approach from Section 2.2, we set the target vector as $z_i = [z_{i,1}, \ldots, z_{i,K}]^\top \in \mathbb{R}^K$ with $z_{i,k} = m_k(\widehat{x}_i, x_i)$, and use $c$ posterior samples $\{\widetilde{x}_i^{(j)}\}_{j=1}^c$ to form the prediction matrix $\widehat{z}_i = [\widehat{z}_{i,1}, \ldots, \widehat{z}_{i,K}]^\top \in \mathbb{R}^{K \times c}$, where $\widehat{z}_{i,k} = [m_k(\widehat{x}_i, \widetilde{x}_i^{(1)}), \ldots, m_k(\widehat{x}_i, \widetilde{x}_i^{(c)})]^\top$.

## 4.2 Multi-task uncertainty quantification

Now consider task-based UQ, as described in Section 2.2 for the case of a single task. In practice, one may want to consider several tasks, such as classifying the presence/absence of several different pathologies from a single image. To extend the task-based UQ method from Section 2.2 to $K$ downstream tasks $\{\mu_k(\cdot)\}_{k=1}^K$, we form the target vector as $z_i = [z_{i,1}, \ldots, z_{i,K}]^\top \in \mathbb{R}^K$ with $z_{i,k} = \mu_k(x_i)$ and use $c$ posterior samples $\{\widetilde{x}_i^{(j)}\}_{j=1}^c$ to form the prediction matrix $\widehat{z}_i = [\widehat{z}_{i,1}, \ldots, \widehat{z}_{i,K}]^\top \in \mathbb{R}^{K \times c}$, where $\widehat{z}_{i,k} = [\mu_k(\widetilde{x}_i^{(1)}), \ldots, \mu_k(\widetilde{x}_i^{(c)})]^\top$.

## 4.3 Multi-round measurement acquisition

In Wen et al. (2024), the authors propose a task-based multi-round measurement protocol where measurements are gradually collected until the conformal interval length falls below a user-specified threshold $\tau$. More precisely, at the end of each measurement round $b \in \{1, \ldots, B\}$, a prediction $\widehat{z}_0^{[b]} \in \mathbb{R}$ and conformal prediction interval $\mathcal{C}^{[b]}(\widehat{z}_0^{[b]})$ are constructed from the cumulative measurements $y_0^{[b]}$. Measurement collection stops if $|\mathcal{C}^{[b]}(\widehat{z}_0^{[b]})| \le \tau$ (or if $b = B$) but otherwise continues to the next round. The goal is to reduce measurement costs while guaranteeing that the collected measurements are sufficient for the task. This is especially useful in applications like accelerated MRI, where long scan times increase both patient discomfort and the likelihood of motion artifacts (Knoll et al., 2020).

A limitation of the multi-round protocol from Wen et al. (2024) is that the marginal coverage guarantee holds for each round in isolation, but not for the multi-round protocol as a whole. In the sequel, we refer to the multi-round protocol from Wen et al. (2024) as the "separate calibration" (SC) method, because it is implemented using a bank of $B$ independently calibrated conformal predictors. That is, although the SC method ensures that $\Pr\{Z_0 \in \mathcal{C}^{[b]}(\widehat{Z}_0^{[b]})\} \ge 1 - \alpha$ for each round $b$ assuming $(Z_0, \widehat{Z}_0^{[b]}), \ldots, (Z_n, \widehat{Z}_n^{[b]})$ are

exchangeable, we really desire that the multi-round coverage

$$P_{\text{multi}} \triangleq \sum_{b=1}^{B} \Pr\left\{Z_0 \in \mathcal{C}^{[b]}(\widehat{Z}_0^{[b]}), \text{final round} = b\right\} \tag{32}$$

$$= \Pr\left\{Z_0 \in \mathcal{C}^{[1]}(\widehat{Z}_0^{[1]}), |\mathcal{C}^{[1]}(\widehat{Z}_0^{[1]})| \leq \tau\right\}$$

$$+ \sum_{b=2}^{B-1} \Pr\left\{Z_0 \in \mathcal{C}^{[b]}(\widehat{Z}_0^{[b]}), |\mathcal{C}^{[b]}(\widehat{Z}_0^{[b]})| \leq \tau, |\mathcal{C}^{[b-1]}(\widehat{Z}_0^{[b-1]})| > \tau, \ldots, |\mathcal{C}^{[1]}(\widehat{Z}_0^{[1]})| > \tau\right\}$$

$$+ \Pr\left\{Z_0 \in \mathcal{C}^{[B]}(\widehat{Z}_0^{[B]}), |\mathcal{C}^{[B-1]}(\widehat{Z}_0^{[B-1]})| > \tau, \ldots, |\mathcal{C}^{[1]}(\widehat{Z}_0^{[1]})| > \tau\right\} \tag{33}$$

is at least $1-\alpha$. This is because, in practice, we don't apriori know which round will be the accepted round, and thus we must ensure coverage in all cases. To address this limitation, we note from (32) that

$$P_{\text{multi}} \geq \sum_{b=1}^{B} \Pr\left\{\cap_{k=1}^{B} Z_0 \in \mathcal{C}^{[k]}(\widehat{Z}_0^{[k]}), \text{final round} = b\right\} = \Pr\left\{\cap_{k=1}^{B} Z_0 \in \mathcal{C}^{[k]}(\widehat{Z}_0^{[k]})\right\}, \tag{34}$$

with equality due to the fact that $\sum_{b=1}^{B} \Pr\{\text{final round} = b\} = 1$. Thus, if

$$\Pr\left\{\cap_{b=1}^{B} Z_0 \in \mathcal{C}^{[b]}(\widehat{Z}_0^{[b]})\right\} \geq 1 - \alpha, \tag{35}$$

then it follows that $P_{\text{multi}} \geq 1-\alpha$. Since (35) is a special case of the joint marginal coverage guarantee (9), we can ensure (35) using multi-target conformal prediction techniques like the one proposed in Section 3.

That said, the above $B$-round protocol handles only scalar $z_0 \in \mathbb{R}$, i.e., a single task. To extend it to $L$ tasks $\{\mu_l(\cdot)\}_{l=1}^{L}$, we set the target vector as $z_i = [z_{i,1}, \ldots, z_{i,K}]^\top \in \mathbb{R}^K$, where $K = BL$ and $z_{i,L(b-1)+l} = \mu_l(x_i)$ for all $b$. Then, for each round $b$, we generate $c$ posterior samples $\{\widetilde{x}_i^{[b](j)}\}_{j=1}^{c}$ via $\widetilde{x}_i^{[b](j)} = g(y_i^{[b]}, \widetilde{v}_i^{[b](j)})$ with i.i.d $\widetilde{v}_i^{[b](j)} \sim \mathcal{N}(0, I)$ and form the predictions $\widehat{z}_i = [\widehat{z}_{i,1}, \ldots, \widehat{z}_{i,K}]^\top \in \mathbb{R}^{K \times c}$ with $\widehat{z}_{i,L(b-1)+l} = [\mu_l(\widetilde{x}_i^{[b](1)}), \ldots, \mu_l(\widetilde{x}_i^{[b](c)})]^\top$. When the joint-coverage guarantee in (9) holds, the prediction intervals for all tasks will simultaneously contain their respective targets with probability at least $1-\alpha$ in the final measurement round.

## 5 Numerical experiments

We now numerically evaluate the proposed asymptotically minimax multi-target conformal prediction method from Section 3.2, along with the independence-assumption (IA)-based method from Messoudi et al. (2020), the quantile-normalization (QN)-based method from Sampson & Chan (2024), and the copula-based "CPTS" method from Sun & Yu (2024), all described in Section 2.3. We compare all four methods using both synthetic data and real-world accelerated-MRI data. For $\widehat{q}_{\frac{\alpha}{2},k}(\cdot)$ and $\widehat{q}_{1-\frac{\alpha}{2},k}(\cdot)$, we use the empirical quantile estimator (6) for all MRI experiments in Section 5.2 but train a quantile regression (Koenker & Bassett, 1978) model for the synthetic experiments in Section 5.1. For the IA method, we use the CQR nonconformity score from (7) with an adjusted error-rate of $\alpha_1 = 1 - (1-\alpha)^{\frac{1}{K}}$ to provide a joint-coverage rate of $1-\alpha$. For the QN method, we compute the nonconformity scores for each target using (14) before taking the max across targets in (12). For the CPTS method, we use the CQR nonconformity score from (7) but otherwise use the implementation settings from Sun & Yu (2024). Since our minimax method can be applied with any nonconformity score $s_k(\cdot, \cdot)$, we test one variation with the CQR score (7), denoted as "CQR+Minimax," and one with the QN score (14), denoted as "QN+Minimax."

For evaluation, we first randomly draw a fixed tuning dataset $d_{\text{tune}}$ from the non-training data. Since the joint guarantee (9) holds over the randomness in the calibration and test data, we perform $T = 10\,000$ Monte Carlo trials, where in each trial $t \in \{1, \ldots, T\}$ we randomly partition the remaining non-training data into a calibration set $d_{\text{cal}}[t]$ with indices $i \in I_{\text{cal}}[t]$ and a test set with indices $i \in I_{\text{test}}[t]$. As a coverage metric, we

evaluate the empirical joint coverage

$$\text{EJC} \triangleq \frac{1}{T} \sum_{t=1}^{T} \frac{1}{|\mathcal{I}_{\text{test}}[t]|} \sum_{i \in \mathcal{I}_{\text{test}}[t]} \prod_{k \in \{1,\dots,K\}} \mathbb{1}\{z_{i,k} \in \mathcal{C}_{\widehat{\lambda}(d_{\text{cal}}[t]),k}(\widehat{z}_i)\}, \tag{36}$$

where $\mathbb{1}\{\cdot\}$ is the indicator function. Since a larger coverage generally indicates a more conservative prediction interval for a given nonconformity score, we also measure the empirical single-target coverage for each target $k$:

$$\text{ESC}_k \triangleq \frac{1}{T} \sum_{t=1}^{T} \frac{1}{|\mathcal{I}_{\text{test}}[t]|} \sum_{i \in \mathcal{I}_{\text{test}}[t]} \mathbb{1}\{z_{i,k} \in \mathcal{C}_{\widehat{\lambda}(d_{\text{cal}}[t]),k}(\widehat{z}_i)\} \tag{37}$$

and, to quantify the corresponding prediction-interval size, we measure the mean interval length:

$$\text{MIL}_k \triangleq \frac{1}{T} \sum_{t=1}^{T} \frac{1}{|\mathcal{I}_{\text{test}}[t]|} \sum_{i \in \mathcal{I}_{\text{test}}[t]} |\mathcal{C}_{\widehat{\lambda}(d_{\text{cal}}[t]),k}(\widehat{z}_i)|. \tag{38}$$

## 5.1 Synthetic data

We start with a synthetic multi-target regression problem. The random data $\{Z_i\}$ are constructed i.i.d across $i \in \{0, 1, \dots, n + n_{\text{tune}} + n_{\text{test}} + n_{\text{train}}\}$ with $Z_i = [Z_{i,1}, Z_{i,2}, Z_{i,3}]^\top$,

$$Z_{i,1} = 10U_i + 10 + \epsilon_{i,1} \tag{39a}$$
$$Z_{i,2} = -2U_i + 1 + \epsilon_{i,2} \tag{39b}$$
$$Z_{i,3} = 0.1U_i^2 + \epsilon_{i,3}, \tag{39c}$$

$U_i \sim \text{Unif}(-5, 5)$, and two cases of random $\epsilon_i = [\epsilon_{i,1}, \epsilon_{i,2}, \epsilon_{i,3}]^\top$. In the first case, $\epsilon_{i,1} \sim \mathcal{N}(10, 1)$, $\epsilon_{i,2} \sim \text{Gamma}(\text{shape} = 1, \text{scale} = 1)$, and $\epsilon_{i,3} \sim \text{Exp}(\text{scale} = 1)$ are independent. In the second case, $\epsilon_i$ is formed by stacking those three independent noise variables into a vector and multiplying by the Cholesky factor of

$$\Sigma = \begin{bmatrix} 1 & 0.8 & 0.7 \\ 0.8 & 1 & 0.4 \\ 0.7 & 0.4 & 1 \end{bmatrix} \tag{40}$$

to construct a correlated noise vector.

In either case, we generate $n_{\text{train}} = 10\,000$ training samples. For each target component $k$, we train two linear quantile regressors $\widehat{q}_{\frac{\alpha}{2},k}(\cdot)$ and $\widehat{q}_{1-\frac{\alpha}{2},k}(\cdot)$ to estimate the $\frac{\alpha}{2}$ and $1-\frac{\alpha}{2}$ quantiles of $Z_{i,k}$, respectively, from $\widehat{z}_i = u_i$. We generate $n_{\text{tune}} = 10\,000$ tuning samples to estimate the empirical CDF for the minimax approach (see (24)) and CPTS approach (see Sun & Yu (2024, Eq.(5))). Then, for each Monte Carlo trial $t$, we generate $n = 10\,000$ calibration samples and $n_{\text{test}} = 5000$ test samples. For fairness, the IA and QN methods use the tuning samples as additional calibration samples.

**Empirical coverage:** Table 1 shows the EJC (36) for each method versus the desired joint coverage $1-\alpha$. There we see that, in the independent-noise case, all four methods perform nearly identically, with EJCs that almost exactly match the desired $1-\alpha$. In the correlated-noise case, however, the IA method yields an overly conservative EJC while the QN, CPTS, and minimax methods give an EJC of almost exactly the desired $1-\alpha$. This behavior is not surprising, since the IA method assumes independent target components while the other methods do not.

**Single-target coverage:** Figure 1(a) and (b) plot the $\max_k \text{ESC}_k$ and $\min_k \text{ESC}_k$ of each method versus the desired joint coverage $1-\alpha$ for the independent- and correlated-noise cases, respectively. With independent noise, the QN and CPTS methods result in the largest ESC spread (i.e., $\max_k \text{ESC}_k - \min_k \text{ESC}_k$), especially at $1-\alpha = 0.8$. With correlated noise, the IA method yields much more conservative $\text{ESC}_k$s than the other methods, while the QN and CPTS methods result in the largest ESC spread, especially for $1-\alpha \le 0.75$.

Table 1: Empirical joint coverage versus desired joint coverage $1-\alpha$ for the synthetic experiment.

| Noise Type | Method | \multicolumn{6}{c}{$1-\alpha$} |
|---|---|---|---|---|---|---|---|
| | | 0.70 | 0.75 | 0.80 | 0.85 | 0.90 | 0.95 |
| Independent | IA (Messoudi et al., 2020) | 0.7000 | 0.7501 | 0.8001 | 0.8501 | 0.9000 | 0.9499 |
| | CPTS (Sun & Yu, 2024) | 0.7002 | 0.7501 | 0.8001 | 0.8501 | 0.9001 | 0.9501 |
| | QN (Sampson & Chan, 2024) | 0.7002 | 0.7501 | 0.8001 | 0.8501 | 0.9000 | 0.9500 |
| | CQR+Minimax (Ours) | 0.7000 | 0.7500 | 0.8000 | 0.8500 | 0.9000 | 0.9501 |
| | QN+Minimax (Ours) | 0.7002 | 0.7502 | 0.8001 | 0.8501 | 0.9000 | 0.9501 |
| Correlated | IA (Messoudi et al., 2020) | 0.7583 | 0.7991 | 0.8393 | 0.8791 | 0.9187 | 0.9584 |
| | CPTS (Sun & Yu, 2024) | 0.7001 | 0.7502 | 0.8001 | 0.8501 | 0.9001 | 0.9502 |
| | QN (Sampson & Chan, 2024) | 0.7001 | 0.7501 | 0.8001 | 0.8500 | 0.9000 | 0.9500 |
| | CQR+Minimax (Ours) | 0.7000 | 0.7501 | 0.8001 | 0.8500 | 0.9000 | 0.9500 |
| | QN+Minimax (Ours) | 0.7001 | 0.7502 | 0.8001 | 0.8501 | 0.9001 | 0.9501 |

**Sensitivity to quantile-estimator quality:** We now investigate the effect of quantile-estimator quality, which can be difficult to guarantee in practice. To do this, we vary the number of training samples $n_{\text{train}}$ used to train the quantile regressor. Since, when $n_{\text{train}}$ is small, the performance can vary significantly over different draws of the training set, we run the experiment five times. Figure 1(c) plots the mean (across these 5 runs) of $\max_k \text{ESC}_k$ and $\min_k \text{ESC}_k$ versus $n_{\text{train}} \in \{50, 500, 1000, 2000\}$ with $\alpha = 0.1$ and correlated noise. The QN method shows a large $\text{ESC}_k$ spread for small values of $n_{\text{train}}$, while the other methods show robustness to $n_{\text{train}}$.

**Sensitivity to tuning samples:** Since the CPTS and minimax methods rely on tuning samples, we now investigate $\text{ESC}_k$ as the number of tuning samples $n_{\text{tune}}$ is varied. (Recall that the IA and QN methods use these tuning samples as additional calibration samples.) Again we run the experiment five times, drawing new training and tuning sets each time. Figure 1(d) plots the mean (across runs) of $\max_k \text{ESC}_k$ and $\min_k \text{ESC}_k$ versus $n_{\text{tune}} \in \{50, 500, 1000, 2000, 5000, 10000\}$ with $\alpha = 0.1$ and correlated noise. As expected, the CPTS and minimax methods perform poorly when $n_{\text{tune}}$ is too small. For sufficiently large $n_{\text{tune}}$, both the CPTS and minimax methods perform on par with QN. But to reach this level of performance, CPTS requires more samples than the minimax methods (in this case, $10\,000$ versus $5000$).

In summary, the IA method is sensitive to noise correlation, while the other methods are not. Meanwhile, the QN method is sensitive to poor quantile estimators (e.g., from too-few training samples) while CPTS and the proposed minimax methods are sensitive to too-few tuning samples. Since conformal prediction is often applied post-hoc, where one has significant control over the tuning/calibration process but no control over the training, CPTS and the proposed minimax methods can be advantageous. We will see this behavior arise in the subsequent MRI experiments, where we have little control over the quantile-estimator performance. Finally, Fig. 1 suggests that the proposed minimax methods offer more balanced single-target coverages than CPTS.

## 5.2 Magnetic resonance imaging

We now compare all five methods on experiments with accelerated magnetic resonance imaging (MRI) (Knoll et al., 2020; Hammernik et al., 2023). MRI is renowned for its ability to provide high-quality soft tissue images without the use of harmful ionizing radiation. However, MRI scans are slow, which compromises patient comfort and throughput and can lead to motion artifacts. In accelerated MRI, the scan time is reduced by a factor of $R$ by collecting only $1/R$ of the measurements required by the Nyquist sampling theorem. Doing so, however, leads to an ill-posed imaging inverse problem, where it is impossible to guarantee recovery of the true image. Thus, for robust diagnoses, uncertainty quantification becomes important.

**Data:** We follow the experimental setup of Wen et al. (2024), which uses the non-fat-suppressed subset of the multicoil fastMRI knee dataset from Zbontar et al. (2018). This subset contains $17\,286$ training images and 2188 validation images. To generate the accelerated measurements, the spatial Fourier domain, known as the "k-space", is retrospectively subsampled with random Cartesian masks at acceleration rates

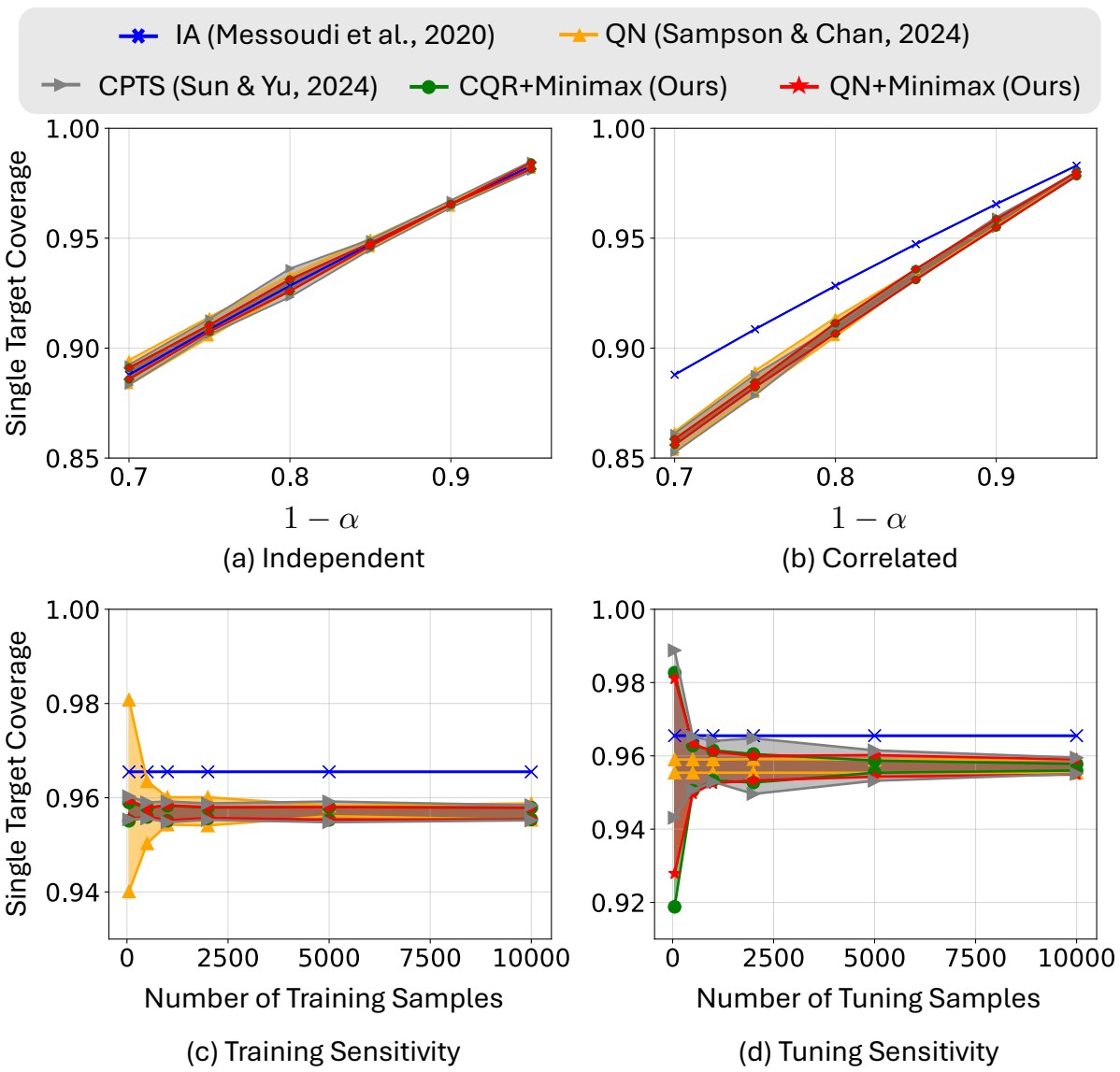

Figure 1: For the synthetic experiment, (a) shows $\min_k \mathrm{ESC}_k$ and $\max_k \mathrm{ESC}_k$ versus desired joint coverage $1-\alpha$ for the independent-noise case and (b) shows the same for the correlated-noise case. Then for the correlated-noise case, (c) shows $\min_k \mathrm{ESC}_k$ and $\max_k \mathrm{ESC}_k$ versus $n_{\mathsf{train}}$ at $1-\alpha = 0.9$ and (d) shows the same versus $n_{\mathsf{tune}}$. The traces in (c) and (d) represent the average across 5 draws of training and tuning data. In some cases the green curves are hidden behind the red curves.

Table 2: Empirical joint coverage versus $1-\alpha$ for the multi-metric and multi-task MRI experiments at $R = 8$.

| Task | Method | $1-\alpha$ | | | | | |
|------|--------|------|------|------|------|------|------|
| | | 0.70 | 0.75 | 0.80 | 0.85 | 0.90 | 0.95 |
| multi-metric | IA (Messoudi et al., 2020) | 0.7797 | 0.8140 | 0.8485 | 0.8856 | 0.9264 | 0.9625 |
| | CPTS (Sun & Yu, 2024) | 0.7025 | 0.7518 | 0.8025 | 0.8525 | 0.9026 | 0.9520 |
| | QN (Sampson & Chan, 2024) | 0.7006 | 0.7502 | 0.8006 | 0.8503 | 0.9005 | 0.9503 |
| | CQR+Minimax (Ours) | 0.7008 | 0.7511 | 0.8005 | 0.8506 | 0.9002 | 0.9506 |
| | QN+Minimax (Ours) | 0.7010 | 0.7502 | 0.8007 | 0.8508 | 0.9002 | 0.9508 |
| multi-task | IA (Messoudi et al., 2020) | 0.7935 | 0.8344 | 0.8677 | 0.8997 | 0.9340 | 0.9697 |
| | CPTS (Sun & Yu, 2024) | 0.7019 | 0.7529 | 0.8034 | 0.8518 | 0.9010 | 0.9526 |
| | QN (Sampson & Chan, 2024) | 0.7005 | 0.7502 | 0.8005 | 0.8502 | 0.9005 | 0.9505 |
| | CQR+Minimax (Ours) | 0.7013 | 0.7506 | 0.8009 | 0.8511 | 0.9004 | 0.9507 |
| | QN+Minimax (Ours) | 0.7013 | 0.7505 | 0.8006 | 0.8509 | 0.9004 | 0.9505 |

$R \in \{16, 8, 4, 2\}$. The masks use Golden Ratio Offset (GRO) sampling (Joshi et al., 2022) and include a fully sampled autocalibration signal (ACS) region in the center, and they are nested such that the measurements collected at each $R$ include all measurements collected at higher $R$. See Wen et al. (2024) for details.

**Models:** For the image-recovery model $f(\cdot)$, we use the popular E2E-VarNet from Sriram et al. (2020), and for the posterior-sampling method $g(\cdot, \cdot)$, we use the conditional normalizing flow (CNF) from Wen et al. (2023) with $c = 32$ posterior samples. Both networks are trained (using the fastMRI training images) to handle acceleration rates $R \in \{16, 8, 4, 2\}$ following the procedure in Wen et al. (2024). The conformal predictors are all given access to the same tuning and calibration samples $\{(\widehat{z}_i, z_i)\}$ and thus the image-recovery and posterior-sampling models are used identically across methods.

**Validation:** We first construct a tuning set $d_{\mathsf{tune}}$ using 656 of the 2188 fastMRI validation samples (i.e., 30%), selected randomly. Since the joint-coverage guarantee (9) holds over the randomness in the calibration and test data, we evaluate performance using $T = 10\,000$ Monte Carlo trials. In each Monte Carlo trial $t \in \{1, \ldots, T\}$, we randomly partition the remaining validation data into a calibration set $d_{\mathsf{cal}}[t]$ of size $n = 1073$ (or 50%) and a test set of size $n_{\mathsf{test}} = 459$ (or 20%) using indices $i \in I_{\mathsf{test}}[t]$. Because the IA and QN methods do not use a tuning set, we add the tuning samples to their calibration sets (now of size $n + n_{\mathsf{tune}} = 1729$) for fair comparison to the CPTS and minimax methods.

### 5.2.1 Multi-metric blind FRIQ assessment

We begin with the multi-metric blind FRIQ assessment problem from Section 4.1. For the FRIQ metrics, we consider PSNR, SSIM, learned perceptual image patch similarity (LPIPS) (Zhang et al., 2018), and deep image structure and texture similarity (DISTS) (Ding et al., 2020). The average FRIQ performance of the E2E-VarNet and CNF on the fastMRI validation set match those reported in Wen et al. (2025).

**Empirical coverage:** Table 2 shows EJC versus desired joint coverage $1-\alpha$ at acceleration $R = 8$. While all methods satisfy the joint-coverage guarantee (9), the EJC of the IA method is overly conservative.

**Single-target coverage:** Figure 2 plots $\max_k \mathrm{ESC}_k$ and $\min_k \mathrm{ESC}_k$ versus $1-\alpha$ at $R = 8$. The QN method suffers from very highly spread $\mathrm{ESC}_k$, reminiscent of the synthetic experiment with poor quantile estimates, while the IA method suffers from overly conservative $\mathrm{ESC}_k$, reminiscent of the synthetic experiment with correlated noise. The CPTS method gives $\max_k \mathrm{ESC}_k$ values similar to IA (i.e., overly conservative) but slightly better for small values of $1-\alpha$, while the CPR-minimax method gives noticeably better $\max_k \mathrm{ESC}_k$ than CPTS, which is consistent with its minimax formulation. (Recall that CPR-minimax and CPTS use the same nonconformity score.) Figure 3 shows $\mathrm{ESC}_k$ individually for each target $k$, revealing that QN provides a massively conservative $\mathrm{ESC}_k \approx 1$ for PSNR and an overly small $\mathrm{ESC}_k$ for LPIPS. Based on Section 5.1, we conjecture that QN's coverage imbalance stems from unreliable quantile estimators $\widehat{q}_{\frac{\alpha}{2}, k}(\cdot)$ and $\widehat{q}_{1-\frac{\alpha}{2}, k}(\cdot)$. When the QN score is used within our minimax framework, however, the single-target coverages become well balanced.

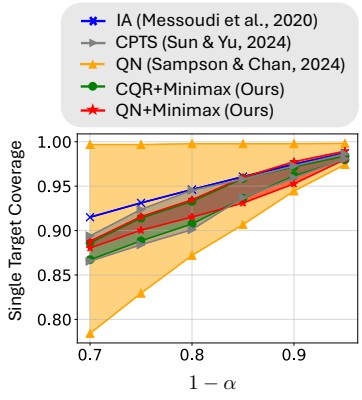

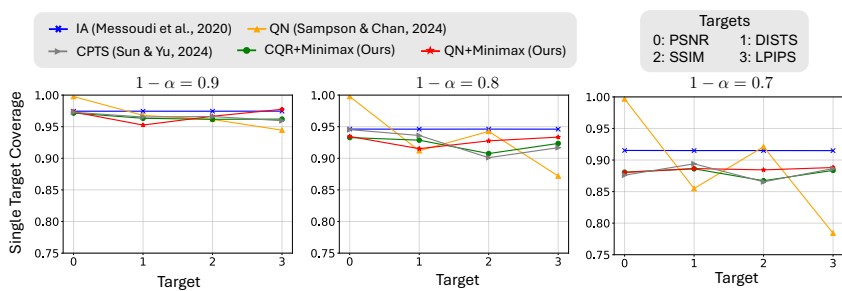

Figure 2: Min and max $\mathrm{ESC}_k$ versus $1-\alpha$ for multi-metric MRI at acceleration $R = 8$.

Figure 3: $\mathrm{ESC}_k$ for each FRIQ metric $k$ and desired coverage level $1-\alpha$ in MRI at acceleration $R = 8$.

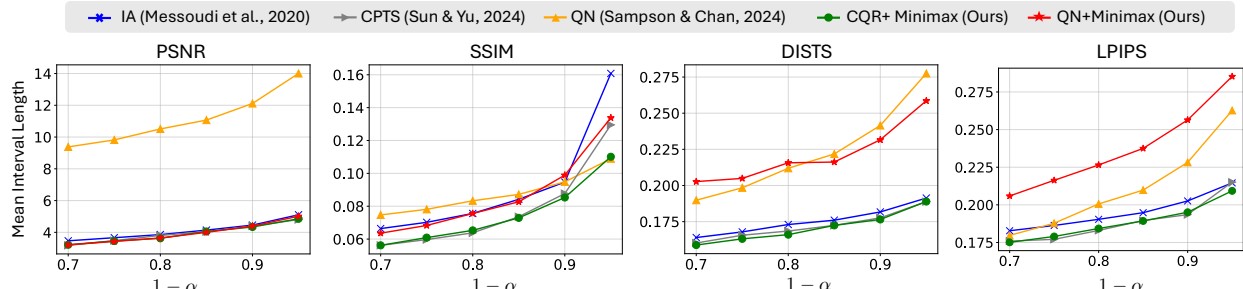

Figure 4: MIL versus $1-\alpha$ for the multi-metric MRI experiments at acceleration $R = 8$.

**Mean interval length:** Figure 4 shows mean interval length $\mathrm{MIL}_k$ versus desired joint coverage $1-\alpha$ for each metric $k$ at $R = 8$. For the PSNR metric, we see QN producing extremely loose prediction intervals at all $1-\alpha$, which is consistent with the overly generous single-target coverage shown in Fig. 3. For the LPIPS metric, although QN gives tighter prediction intervals than QN+Minimax, they come at the cost of very weak single-target coverage guarantees, as shown in Fig. 3. Although the IA method performs very consistently across the four target metrics, both CPTS and the proposed CQR+Minimax give tighter predictions intervals in every instance. Although CQR+Minimax performs similarly to CPTS in most cases, it gives noticeably tighter prediction intervals for SSIM and LPIPS at high $1-\alpha$.

Overall, we see the minimax methods behaving as expected in this multi-metric MRI experiment, i.e., providing well-balanced single-target coverages and thereby avoiding overly large prediction intervals.

### 5.2.2 Multi-task uncertainty quantification

We now consider a multi-task UQ problem, where the goal is to ascertain the presence/absence of each of $K = 5$ different pathologies in accelerated MRI with $R = 8$. We assume that a multi-label soft-output classifier $\mu(\cdot)$ has been trained on clean images $x_i$ to output a vector of probabilities $z_i \in [0,1]^K$. At inference time, since we have access to only the accelerated measurements $y_0$ and not the true image $x_0$, the goal is to construct, for each pathology $k$, a prediction interval $\mathcal{C}_{\widehat{\lambda}(d_{\mathrm{cal}}),k}(\widehat{z}_0)$ that contains the true soft-output $z_{0,k} = [\mu(x_0)]_k$ with some probabilistic guarantee. We apply the IA, QN, CPTS, CQR+Minimax, and QN+Minimax methods as described in Section 4.2.

For the classifier, we use a ResNet-50 (He et al., 2016) with $K = 5$ outputs in the final linear layer. To train it, we first initialize using ImageNet weights, then pretrain using SimCLR loss (Chen et al., 2020) on the (unlabeled) fastMRI knee data, and lastly fine-tune using binary cross-entropy loss on the (labeled) fastMRI+

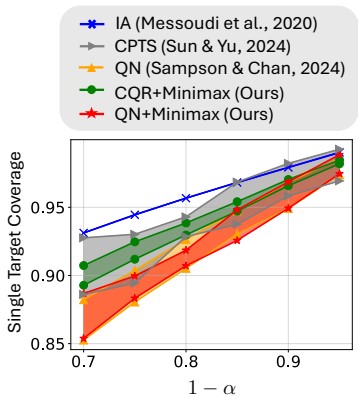

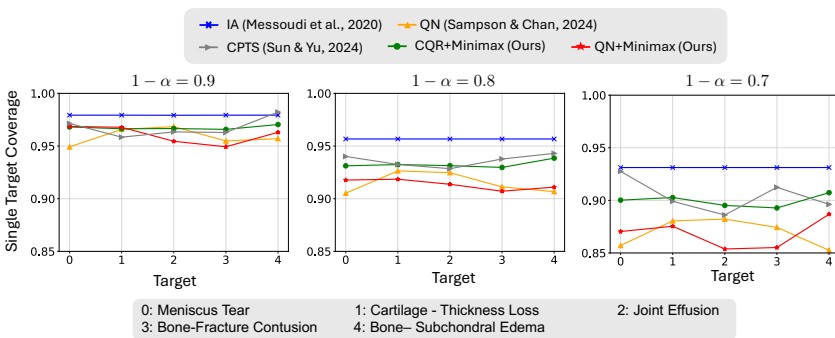

Figure 5: Min and max $\text{ESC}_k$ versus $1-\alpha$ for multi-task MRI at acceleration $R = 8$.

Figure 6: $\text{ESC}_k$ versus target $k$ for several values of $1-\alpha$ in multi-task MRI at acceleration $R = 8$.

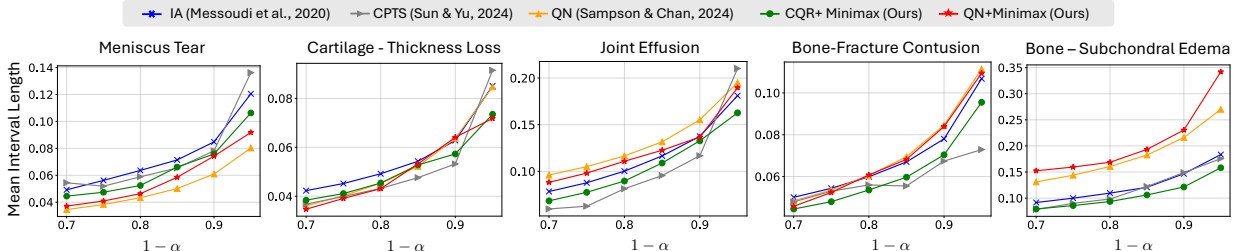

Figure 7: $\text{MIL}_k$ versus $1-\alpha$ for each task $k$ in multi-task MRI at acceleration $R = 8$.

knee data (Zhao et al., 2022). The chosen pathologies, listed in Fig. 6, were the five with the most fastMRI+ samples. Additional details can be found in Appendix B.

**Empirical coverage:** Table 2 reports EJC versus desired joint coverage $1-\alpha$. There we see that all methods satisfy the joint-coverage guarantee (9), but that the EJC of IA is overly conservative. In fact, the IA method is even more conservative in this multi-task experiment than it was in the multi-metric experiment, which (recalling Section 5.1) may be due to increased correlation among the non-conformity scores for different tasks.

**Single-target coverage:** Figure 5 plots the $\max_k \text{ESC}_k$ and $\min_k \text{ESC}_k$ versus $1-\alpha$ for each method at $R = 8$. Again, the IA method gives overly conservative $\text{ESC}_k$ for all $1-\alpha$. The CPTS method suffers from imbalanced $\text{ESC}_k$ across targets, and consequently its $\max_k \text{ESC}_k$ is similar to IA's in most cases. By contrast, the CQR+Minimax method yields well-balanced $\text{ESC}_k$ over the entire range. As for the methods that use the QN nonconformity score, we see that QN and QN+Minimax give similar $\text{ESC}_k$ except around $1-\alpha = 0.8$, where QN+Minimax is better balanced. Figure 6 shows $\text{ESC}_k$ individually for each target $k$ and shows that each method distributes coverage across targets in a unique manner, especially at $1-\alpha = 0.7$.

**Mean interval length:** Figure 7 plots $\text{MIL}_k$ versus desired joint coverage $1-\alpha$ for each target $k$. Among the CQR-based methods (i.e., IA, CPTS, CQR+Minimax), we see that IA tends to give the loosest prediction intervals, while CPTS and CQR+Minimax trade for the tightest interval on a case-by-case basis. For large $1-\alpha$, however, CPTS gives unnecessarily loose intervals for 3 of the 5 classes, while CQR+Minimax avoids this unwanted behavior. Relative to the CQR-based methods, the QN-based ones are much less consistent, in that they give very tight intervals for some classes (e.g., Meniscus Tear) but very loose ones for others (e.g., Bone–Subchondral Edema). Overall, QN and QN+Minimax achieve similar performance, with one or the other winning on a case-by-case basis. The loose interval produced by QN+Minimax at $1-\alpha = 0.95$ is most likely due to the use of relatively few tuning samples (recall Fig. 1(d)), recalling that the proposed methods are minimax only asymptotically.

### 5.2.3 Multi-round measurement acquisition with FRIQ guarantees

We now consider applying the multi-round measurement protocol from Section 4.3 to accelerate MRI while providing a probabilistic FRIQ guarantee. We adopt the experimental setup of Wen et al. (2025), where measurements are collected over $B = 5$ rounds at acceleration rates $R \in \{16, 8, 4, 2, 1\}$ but stop as soon as a conformal upper bound on DISTS[1] falls below a threshold of[2] $\tau = 0.16$. To adapt the proposed multi-target method from Section 4.3 to this upper-bounding setup, we run the IA, CPTS, and CQR+Minimax methods with the one-sided CQR nonconformity score $s_k(\widehat{z}_{i,k}, z_{i,k}) = z_{i,k} - \widehat{q}_{1-\frac{\alpha}{2}}(\widehat{z}_{i,k})$ and we run QN and QN+Minimax with one-sided QN nonconformity score

$$s_k(\widehat{z}_{i,k}, z_{i,k}) = \left(z_{i,k} - \widehat{q}_{1-\frac{\alpha}{2},k}(\widehat{z}_i)\right) \frac{\widehat{q}_{1-\frac{\alpha}{2},1}(\widehat{z}_i) - \widehat{q}_{\frac{\alpha}{2},1}(\widehat{z}_i)}{\widehat{q}_{1-\frac{\alpha}{2},k}(\widehat{z}_i) - \widehat{q}_{\frac{\alpha}{2},k}(\widehat{z}_i)}, \tag{41}$$

as motivated by (14).

Figure 8(a) plots the empirical accepted coverage

$$\text{EAC} \triangleq \frac{1}{T} \sum_{t=1}^{T} \frac{1}{|\mathcal{I}_{\text{test}}[t]|} \sum_{i \in \mathcal{I}_{\text{test}}[t]} \mathbb{1}\{z_i \in \mathcal{C}^{[b_i]}(\widehat{z}_i^{[b_i]})\}, \tag{42}$$

where $b_i$ denotes the accepted round for the $i$th sample, versus the desired accepted coverage of $1-\alpha$ for the IA, QN, CPTS, CQR+Minimax, and QN+Minimax versions of the multi-target method proposed in Section 4.3, as well as the separate-calibration (SC) method from Wen et al. (2024) discussed in Section 4.3. The figure shows that the measurements accepted by the SC method do not provide the desired coverage, while those accepted by the multi-target methods do, validating the goal of Section 4.3. Figure 8(b) plots the average accepted acceleration-rate

$$R_{\text{avg}} \triangleq \left(\frac{1}{T} \sum_{t=1}^{T} \frac{1}{|\mathcal{I}_{\text{test}}[t]|} \sum_{i \in \mathcal{I}_{\text{test}}[t]} \frac{1}{R_{b_i}}\right)^{-1} \tag{43}$$

versus $1-\alpha$, where $R_{b_i}$ is the acceleration rate at the accepted round for test sample $i$. Although the SC method achieves higher $R_{\text{avg}}$ than the multi-target methods, it comes at the cost of not providing a coverage guarantee on accepted samples. Among the multi-target methods, the CPTS and CQR+Minimax methods yield the highest $R_{\text{avg}}$ in all cases, demonstrating the advantage of tight conformal bounds. However, the much lower computational complexity of CQR+Minimax makes it advantageous in practice.

### 5.2.4 Multi-round measurement acquisition with downstream classification guarantees

Finally, we consider an application of the multi-round measurement protocol from Section 4.3 that aims to accelerate MRI while providing a probabilistic guarantee on downstream classification of multiple pathologies. For this we combine the multi-round setup from Section 5.2.3 with the multi-label setup from Section 5.2.2. In particular, we take measurements over $B = 5$ rounds at acceleration rates $R \in \{16, 8, 4, 2, 1\}$ but stop as soon as the prediction interval lengths for all $L$ pathology labels fall below[3] $\tau = 0.1$. Rather than exclusively considering $L = 5$ pathology labels, we experiment with $L \in \{2, 3, 4, 5\}$ to investigate the effect of $L$. Here, $L = l$ corresponds to the $l$ most prevalent classes in the fastMRI+ training data.

For a desired accepted coverage of $1-\alpha = 0.9$, Fig. 9(a) plots EAC versus the number of labels $L$. There we see that all methods meet the desired coverage for all $L$, which shows that the multi-round methodology proposed in Section 4.3 is flexible with regards to the choice of conformal predictor. Figure 9(b) plots $R_{\text{avg}}$ versus the number of labels $L$ for different conformal predictors. This figure shows that the proposed CQR+Minimax method yields the highest $R_{\text{avg}}$ across all tested values of $L$. It also shows that $R_{\text{avg}}$ decreases with $L$, which is expected because the stopping criterion becomes more strict with larger $L$.

---

[1]A recent clinical MRI study (Kastryulin et al., 2023) evaluated 35 FRIQ metrics and found that DISTS correlated best with radiologists' ratings of perceived noise level, contrast level, and artifacts when comparing reconstructions to ground-truth images.

[2]Wen et al. (2025) chose $\tau = 0.16$ because this is the DISTS threshold at which a single-round measurement scheme at acceleration $R = 2$ achieves $1-\alpha = 0.95$ coverage.

[3]We chose $\tau = 0.1$ purely for the sake of demonstration. A practically meaningful $\tau$ could be attained with expert guidance through clinical trials.

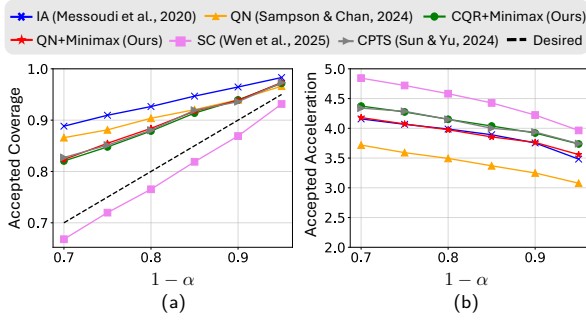
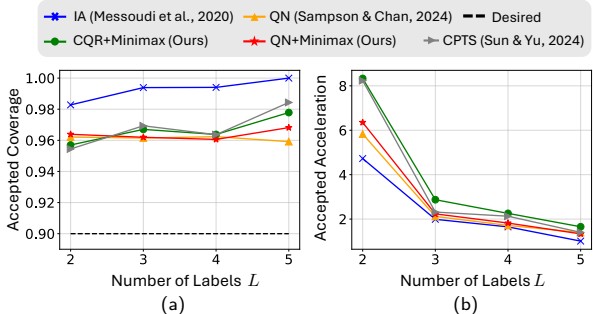

Figure 8: For multi-round measurements that stop as soon as the DISTS upper bound falls below $\tau = 0.16$, (a) plots EAC and (b) plots $R_{\text{avg}}$ versus the desired accepted coverage $1-\alpha$.

Figure 9: For multi-round measurements that stop as soon as the prediction intervals for all pathology labels fall below $\tau = 0.1$, (a) plots EAC and (b) plots $R_{\text{avg}}$ versus the number of labels $L$ at $1-\alpha = 0.9$.

## 6 Conclusion

Motivated by the need for multi-target uncertainty quantification in imaging inverse problems, we propose a minimax-based approach to multi-target conformal prediction. The proposed method aims to minimize the maximum single-target coverage, $\max_k \Pr\{Z_{0,k} \in \mathcal{C}_{\widehat{\lambda}(D_{\text{cal}}),k}(\widehat{Z}_0)\}$, across targets $k$ subject to a marginal joint-coverage guarantee of the form $\Pr\{\cap_{k=1}^K Z_{0,k} \in \mathcal{C}_{\widehat{\lambda}(D_{\text{cal}}),k}(\widehat{Z}_0)\} \geq 1-\alpha$, where $\alpha$ is user-specified. Because our approach is an instance of split conformal prediction, it guarantees marginal joint-coverage with finite-sized tuning and calibration datasets under the usual test/calibration exchangeability condition. Furthermore, it converges to the minimax solution as the size of the tuning and calibration sets grow to infinity. In addition to our minimax multi-target conformal predictor, we propose a multi-round measurement acquisition scheme that guarantees marginal coverage of the final-round prediction interval. We numerically compared the proposed minimax multi-target predictor to several existing methods on a synthetic-data problem as well as four accelerated-MRI problems and found that the proposed minimax method gives better balanced single-target coverages while guaranteeing joint marginal coverage. In addition, we numerically investigated the proposed multi-round measurement scheme and confirmed that it provides marginal accepted coverage when used with a variety of conformal predictors.

### Limitations

There are several limitations to this work. First, like with many conformal prediction methods, the joint-coverage guarantee (9) holds only for prediction/target pairs $(\widehat{Z}_i, Z_i)$ that are statistically exchangeable over the test and calibration data. Furthermore, to prove that our approach is asymptotically minimax, we assumed that the nonconformity scores $\{S_{i,k}\}_{i=1}^{n+n_{\text{tune}}}$ are i.i.d. Further work is needed to generalize these restrictions, and the works Tibshirani et al. (2019), Barber et al. (2023), Cauchois et al. (2024) suggest modifications that address non-exchangeability. In addition, the proposed applications to MRI are preliminary, in that rigorous clinical trials are needed before they are adopted in practice.

### Broader impact statement

We expect that our methodology will positively impact the field of imaging inverse problems by providing prediction intervals on multiple estimation targets that involve the (unknown) true image. These intervals inform the practitioner of how much uncertainty the measurement-and-reconstruction process introduces to downstream tasks, and whether the collected measurements are sufficient for a given reconstruction method. Furthermore, the proposed multi-round acquisition protocol allows one to collect fewer measurements while still providing guarantees on estimation performance. However, clinicians must be careful when interpreting the results, understanding, for example, that our coverage guarantees are marginal and not conditional. As such, they hold only when averaged over many different test samples and calibration sets, rather than for a

specific test sample and/or calibration set. Furthermore, they hold only when the test sample is statistically exchangeable with the calibration samples.

## Acknowledgments

This work was supported in part by the National Institutes of Health under Grant R01-EB029957.

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

# A Proof of Theorem 2

In this section, we show that $\widehat{\lambda}(d_{\mathsf{cal}})$ in (27) converges to $\widehat{\lambda}$ in (23) as $n \to \infty$ and $n_{\mathsf{tune}} \to \infty$. We first recall the definition of almost-sure convergence.

**Definition 1** (Almost-sure convergence). *Let $(X_n)_{n \geq 1}$ be a sequence of random variables defined on a probability space $(\Omega, \mathcal{F}, P)$. We say that $X_n$ **converges almost surely** (or with probability 1) to a random variable $X$, denoted as $X_n \xrightarrow{a.s.} X$, if*

$$\Pr\left\{ \lim_{n \to \infty} X_n = X \right\} = 1.$$

*That is, the outcomes $\omega \in \Omega$ under which $X_n(\omega)$ converges to $X(\omega)$ occur with probability one.*

We now state two theorems that form the basis for our convergence analysis.

**Theorem 3** (Glivenko-Cantelli (Fristedt & Gray, 2013)). *Suppose $X_1, \ldots, X_n$ are i.i.d random variables with CDF $F(\cdot)$. Define the empirical CDF as*

$$\widehat{F}(x) \triangleq \frac{|\{i : X_i \leq x, \ i = 1, \ldots, n\}|}{n}.$$

*Then $\widehat{F}(\cdot)$ converges uniformly to $F(\cdot)$ almost surely, i.e.*

$$\sup_{x \in \mathbb{R}} |\widehat{F}(x) - F(x)| \xrightarrow{a.s.} 0.$$

**Theorem 4.** *Let $X_1, X_2, \ldots, X_n$ be i.i.d random variables with CDF $F(\cdot)$. Define the quantile at level $p \in (0,1)$ as*

$$Q(p) = \inf\{x : F(x) \geq p\}$$

*and the empirical quantile at level $p \in (0,1)$ as*

$$Q_n(p) = \inf\{x : \widehat{F}(x) \geq p\},$$

*where $\widehat{F}(\cdot)$ is the empirical CDF. For a fixed level $p$, construct the $n$-dependent level*

$$\gamma_n = \frac{\lceil p(n+1) \rceil}{n},$$

*which approaches $p$ as $n \to \infty$. If $F(\cdot)$ is continuous and strictly increasing at $Q(p)$, then*

$$Q_n(\gamma_n) \xrightarrow{a.s} Q(p).$$

*That is, the empirical quantile at level $\gamma_n$ converges almost surely to the true quantile at level $p$.*

*Proof.* First, we analyze the convergence of $\gamma_n$. Observe that

$$\gamma_n = \frac{\lceil p(n+1) \rceil}{n} = \frac{p(n+1) + \Delta_n}{n} = p + \frac{p + \Delta_n}{n} \tag{44}$$

where $\Delta_n \in [0, 1)$ accounts for the rounding of the ceiling function. Thus

$$p < \gamma_n < p + \frac{2}{n}, \tag{45}$$

and $\lim_{n \to \infty} \gamma_n = p$. Next, we look to bound $Q_n(\gamma_n)$ as $n \to \infty$. For any fixed $\epsilon > 0$, and assuming $F(\cdot)$ is continuous and strictly increasing at $Q(p)$, we have

$$F(Q(p) - \epsilon) < p < F(Q(p) + \epsilon).$$

And by Theorem 3, the Glivenko-Cantelli theorem, $\widehat{F}(\cdot)$ converges uniformly to $F(\cdot)$ almost surely. This means that, for any $\delta > 0$, there almost surely exists an $N$ such that, for all $n \geq N$ and for all $x \in \mathbb{R}$,

$$|\widehat{F}(x) - F(x)| \leq \delta.$$

By choosing

$$\delta < \min\{p - F(Q(p) - \epsilon), F(Q(p) + \epsilon) - p\}$$

we get

$$F(Q(p) - \epsilon) + \delta < p < F(Q(p) + \epsilon) - \delta, \tag{46}$$

and so

$$\widehat{F}(Q(p) - \epsilon) < p < \widehat{F}(Q(p) + \epsilon) \tag{47}$$

for all $n \geq N$. We now establish two intermediate results.

**Lemma 5.** *For sufficiently large $n$, we have $Q_n(\gamma_n) \geq Q(p) - \epsilon$.*

*Proof.* We prove the claim using contradiction. Suppose that $Q_n(\gamma_n) < Q(p) - \epsilon$. Then, due to the non-decreasing property of $\widehat{F}(\cdot)$, we have

$$\widehat{F}(Q_n(\gamma_n)) \leq \widehat{F}(Q(p) - \epsilon) \tag{48}$$

for any $n$. Furthermore, since $\widehat{F}(Q_n(\gamma_n)) \geq \gamma_n$ for any $n$ by the definition of the empirical quantile, and since $\gamma_n > p$ from (45), we have

$$\widehat{F}(Q(p) - \epsilon) \geq \gamma_n > p. \tag{49}$$

However, (49) contradicts (47) when $n \geq N$. This implies that $Q_n(\gamma_n) \geq Q(p) - \epsilon$ for sufficiently large $n$. ∎

**Lemma 6.** *For sufficiently large $n$, we have $Q_n(\gamma_n) \leq Q(p) + \epsilon$.*

*Proof.* We prove the claim using contradiction. Suppose that $Q_n(\gamma_n) > Q(p) + \epsilon$. Recall that, by definition, $Q_n(\gamma_n) = \inf\{x : \widehat{F}(x) \geq \gamma_n\}$. Thus if $Q(p) + \epsilon < Q_n(\gamma_n)$ then

$$\widehat{F}(Q(p) + \epsilon) < \gamma_n. \tag{50}$$

And recall from (46) that $F(Q(p) + \epsilon) - \delta > p$, or equivalently that

$$F(Q(p) + \epsilon) - \frac{\delta}{2} > p + \frac{\delta}{2}. \tag{51}$$

From Theorem 3, the Glivenko-Cantelli theorem, $\widehat{F}(\cdot)$ converges uniformly to $F(\cdot)$ almost surely. This means that, for the given $\delta$, there almost surely exists an $N'$ such that, for all $n \geq N'$ and any $x \in \mathbb{R}$,

$$|\widehat{F}(x) - F(x)| \leq \frac{\delta}{2} \quad \Rightarrow \quad \widehat{F}(x) \geq F(x) - \frac{\delta}{2}. \tag{52}$$

Combining (51) and (52), we have that, for all $n \geq N'$,

$$\widehat{F}(Q(p) + \epsilon) \geq F(Q(p) + \epsilon) - \frac{\delta}{2} > p + \frac{\delta}{2}.$$

From (45), we see that, for $n \geq 4/\delta$,

$$\gamma_n < p + \frac{2}{n} \leq p + \frac{\delta}{2}.$$

Thus, for sufficiently large $n$, we have

$$\widehat{F}(Q(p) + \epsilon) > \gamma_n,$$

which contradicts (50). This implies that $Q_n(\gamma_n) \leq Q(p) + \epsilon$ for large $n$. ∎

Lemma 5 and Lemma 6 hold almost surely for an arbitrary $\epsilon > 0$, and together say that

$$Q(p) - \epsilon \leq Q_n(\gamma_n) \leq Q(p) + \epsilon$$

for sufficiently large $n$. Since we can make $\epsilon$ arbitrarily small, we have that

$$\lim_{n \to \infty} Q_n(\gamma_n) = Q(p),$$

almost surely, and thus $Q_n(\gamma_n) \xrightarrow{a.s.} Q(p)$. $\qquad\square$

Having established Theorem 3 and Theorem 4, we now return to our main objective, which is proving that the $\widehat{\lambda}(d_{\sf cal})$ in (27) converges to the $\widehat{\lambda}$ in (23). For clarity, we restate Theorem 2 here.

**Theorem** (Restatement of Theorem 2). *For each target component $k = 1, \ldots, K$, suppose that the non-conformity scores $\{S_{i,k}\}_{i=1}^{n+n_{\sf tune}}$ are i.i.d with CDF $F_{S_k}(\cdot)$, and for $T \triangleq \max_k F_{S_k}(S_k)$, suppose that $F_T(\cdot)$ is continuous and strictly increasing at the $(1-\alpha)$-level quantile of $T$. Then $\widehat{\lambda}(d_{\sf cal})$ from (27) converges to $\widehat{\lambda}$ from (23) almost surely as $n \to \infty$ and $n_{\sf tune} \to \infty$.*

*Proof.* We first analyze the effect of $n_{\sf tune} \to \infty$ for an arbitrary fixed $n$. Recall that the empirical CDF $\widehat{F}_{S_k}(\cdot)$ of the nonconformity score for the $k$th component is computed as in (24) using the tuning samples $\{S_{i,k}\}_{i=n+1}^{n+n_{\sf tune}}$. From Theorem 3, $\widehat{F}_{S_k}(\cdot)$ converges uniformly to the CDF $F_{S_k}(\cdot)$ almost surely as $n_{\sf tune} \to \infty$. As a result, it follows that for each *calibration* nonconformity score $S_{i,k}$, where $i \in \{1, \ldots, n\}$, we have

$$\overline{S}_{i,k} \triangleq \widehat{F}_{S_k}(S_{i,k}) \xrightarrow{a.s.} F_{S_k}(S_{i,k})$$

as $n_{\sf tune} \to \infty$, recalling the definition of the transformed score $\overline{S}_{i,k}$ from (25). Let us now consider the maximum transformed score $\overline{S}_i \triangleq \max_k \{\overline{S}_{i,k}\}_{k=1}^K$ defined in (26). Since the maximum function is continuous everywhere on $\mathbb{R}^K$ and $\widehat{F}_{S_k}(S_{i,k}) \xrightarrow{a.s.} F_{S_k}(S_{i,k})$, the continuous mapping theorem implies that

$$\overline{S}_i = \max_k \widehat{F}_{S_k}(S_{i,k}) \xrightarrow{a.s.} \max_k F_{S_k}(S_{i,k}) \triangleq T_i$$

as $n_{\sf tune} \to \infty$. Because $\{S_{i,k}\}_{i=1}^n$ are assumed to be i.i.d with CDF $F_{S_k}(\cdot)$, we see that $\{T_i\}_{i=1}^n$ are i.i.d with CDF $F_T(\cdot)$ for $T \triangleq \max_k F_{S_k}(S_k)$.

Next, we analyze the effect of $n \to \infty$. Let us denote the $n$-sample empirical quantile of $T$ as $Q_n(\cdot)$ and the quantile of $T$ as $Q(\cdot)$. Recall from (27) that

$$\widehat{\lambda}(D_{\sf cal}) = Q_n\left( \frac{\lceil (1-\alpha)(n+1) \rceil}{n} \right).$$

Because $F_T(\cdot)$ is assumed to be continuous and strictly increasing at $Q(1-\alpha)$, Theorem 4 establishes that, as $n \to \infty$,

$$\widehat{\lambda}(D_{\sf cal}) = Q_n\left( \frac{\lceil (1-\alpha)(n+1) \rceil}{n} \right) \xrightarrow{a.s.} Q(1-\alpha). \tag{53}$$

Finally, recall the definition of $\widehat{\lambda}$ from (23):

$$\widehat{\lambda} = \arg\min_\lambda \lambda \quad \text{s.t.} \ \ \Pr\{\cap_{k=1}^K F_{S_k}(S_k) \leq \lambda\} \geq 1 - \alpha.$$

The constraint can be rewritten as

$$\Pr\{\max_k F_{S_k}(S_k) \leq \lambda\} = \Pr\{T \leq \lambda\} \geq 1 - \alpha,$$

which allows (23) to be rewritten as

$$\widehat{\lambda} = \arg\min_\lambda \lambda \quad \text{s.t.} \ \ \Pr\{T \leq \lambda\} \geq 1 - \alpha. \tag{54}$$

Table 3: Number of positive samples in the non-fat-suppressed subset of the fastMRI+ knee dataset.

| Label | Positive Training Samples | Positive Validation Samples |
|---|---|---|
| Meniscus Tear | 1921 | 335 |
| Cartilage - Partial Thickness loss/defect | 871 | 176 |
| Joint Effusion | 225 | 41 |
| Bone-Fracture/Contusion/dislocation | 97 | 6 |
| Bone - Subchondral edema | 76 | 21 |

Table 4: Classifier performance on the fastMRI+ validation set.

| Label | Accuracy | Precision | Recall | AUROC |
|---|---|---|---|---|
| Meniscus Tear | 0.6595 | 0.3005 | 0.9784 | 0.889 |
| Cartilage - Partial Thickness loss/defect | 0.6184 | 0.1558 | 0.8988 | 0.8564 |
| Joint Effusion | 0.9031 | 0.1356 | 0.8000 | 0.9465 |
| Bone-Fracture/Contusion/dislocation | 0.7715 | 0.0060 | 0.5000 | 0.7971 |
| Bone - Subchondral edema | 0.5704 | 0.0127 | 0.5714 | 0.6338 |
| Average | 0.7046 | 0.1221 | 0.7497 | 0.8246 |

But the $\widehat{\lambda}$ in (54) is simply the $(1 - \alpha)$-level quantile of $T$. In other words,

$$\widehat{\lambda} = \inf\{\lambda : F_T(\lambda) \geq 1 - \alpha\} = Q(1 - \alpha). \tag{55}$$

Finally, combining (53) with (55), we conclude that

$$\widehat{\lambda}(D_{\mathsf{cal}}) \xrightarrow{a.s.} \widehat{\lambda}$$

as $n_{\mathsf{tune}} \to \infty$ and $n \to \infty$. $\qquad\square$

## B   Classifier Details

We train the multi-label classifier on the $K = 5$ labels with the most annotations in the non-fat-suppressed subset of the fastMRI+ knee data from Zhao et al. (2022). Table 3 shows the number of positive samples for each of those labels. Note that images with multiple instances of the same pathology only count as a single positive sample.

We implement and train the multi-label classifier using nearly the same procedure as Wen et al. (2024). In particular, we start by initializing a standard ResNet-50 (He et al., 2016) with the pretrained ImageNet weights from (Deng et al., 2009), after which we reduce the number of final-layer outputs to $K = 5$. Then we pretrain the network in a self-supervised fashion using the (unlabeled) non-fat-suppressed fastMRI knee data following the SimCLR procedure from Chen et al. (2020) with a learning rate of 0.0002, batch size of 128, and 500 epochs. Finally, we perform supervised fine-tuning using binary cross-entropy loss on the fastMRI+ data, where we address class imbalance by weighting the loss contribution from each class by the ratio of negative labels to positive labels for that particular class. To encourage adversarial robustness, we use the same $l_2$-bounded gradient ascent attack as Wen et al. (2024), and we train the classifier for 150 epochs with a batch size of 128, learning rate of 5e−5, and weight decay of 1e−7. Finally, we save the model checkpoint with the lowest validation loss. Performance on the validation dataset is shown in Table 4.

