# OpenReview forum: "Minimax Multi-Target Conformal Prediction with Applications to Imaging Inverse Problems"
_TMLR — Accepted by TMLR_

### Review · Reviewer_kgLm · 2025-07-28

**Summary Of Contributions:**

The paper presents a new conformal prediction method for multi-target settings, which ensures simultaneously valid prediction sets for one or more real-valued targets, and tests this method in the context of accelerated MRI experiments. The method stems from a minimax formulation, which essentially finds the best prediction interval parameter lambda out of those that ensure joint distribution free coverage in every dimension of the target vector. Expected validity guarantees, including provable coverage asymptotically and in finite samples, are provided under standard assumptions. Aside from single-shot settings (such as measuring multiple metrics on a recovered image), this method is also applied to provide coverage in the context of multi-round "measurement acquisition" schemes. The experiments showcase the improvements in metrics such as coverage rates and prediction interval length in each dimension of the target.

**Audience:**

Yes

**Claims And Evidence:**

Yes

**Requested Changes:**

As per the above, here are the main questions that I consider important to address:

1. Please introduce a detailed discussion comparing to the Learn-then-Test approach and resulting methods based on multiple hypothesis testing rather than the conformal approach. In particular, I'd like for such a comparison to happen from the (a) methodological/expository point of view (i.e. a careful discussion of how LTT achieves its guarantees vs. the proposed method, and in particular a discussion for the readers about the difference in provided guarantees), and (b) experimental point of view (i.e. does the proposed method achieve less conservative coverage/length than LTT?).

2. Empirically, I would like to request extra visualizations that would (a) better showcase the tradeoffs induced by the multiple objectives, to better understand how the different methods that are compared deal with them; and (b) compare the methods on another task, preferably a well-designed synthetic one, in particular to better delineate the limitations of the proposed method due to its conservatism.

3. (Less pressing than above two, but still quite desirable) A more focused discussion of the minimax nature of the method as per the last of the Weaknesses; this would help the readers navigate (a) the interesting contrast between a robust nature of the proposed method vs. its more efficient performance than the alternatives; and (b) understand future work avenues, such as those stemming from replacing max -> softmax.

**Strengths And Weaknesses:**

Strengths:

+ The proposed method, as showcased in the paper, obtains more robust/better performance than existing conformal-type methods in the field of medical imaging, according to several success metrics (including a domain specific one --- namely better achieved acceleration in MRI).

+ The proposal is a natural one from the minimax perspective, and a simple one --- it obtains a bounding box in higher dimensions, foregoing more sophisticated and potentially less robust/interpretable approaches with different geometries (ellipsoidal etc.). (More contrived prior approaches include the Sampson and Chan (2024) method with an extra quantile-regression-based score renormalization step, which this paper includes in the discussion/evals.)

Weaknesses:

- While the proposed approach is well-situated by the authors in the context of conformal prediction based tools, that is not so with respect to the slightly more expansive toolkit of distribution-free UQ. In particular, in the multi-target context, the well-known Learn then Test (LTT) method of Angelopoulos et al pioneered a hypothesis-testing framework that helps obtain guarantees that can simultaneously bound multiple notions of (monotonic or nonmonotonic) risk (not restricted to, but including, miscoverage rates). LTT is also in particular well-established in the empirical terrain, including CV applications. Thus, this appears to be a substantial omission.

- The empirics, while conducted in a (from what I can tell) well-established and motivated medical imaging setting, lack a certain amount of details. For instance, the Sampson and Chan QN method, that is directly compared to, aims to combine the scores into one (similar to the presented method), but they go through the more convoluted step of renormalizing the scores using quantile regression. Their point is to mitigate the disparity between the different dimensions' scores' magnitudes, which at first glance their formula manages to achieve in some sense. However, the minimax approach indeed achieves (at least somewhat) better performance in presented experiments, which makes one wonder: Is this due to the QN method being unable to handle the differing scale or distribution of the different metrics' values?

    ---> For this and other typical multi-objective reasons, I would have liked to see more visualizations of the different score dimensions, to help the reader better understand the trade-offs that the tested methods have to navigate.

    ---> In a similar vein, I would have liked to see some well-thought-out synthetic experiments on the proposed method comparing it to prior approaches, to better understand: to what extent is the proposed method better than the others due to some special structure in medical imaging (inverse) tasks, and to what extent would that advantage erode under different relative distributions of the different dimensions' scores? In particular, the method doesn't directly control the interval length, and therefore it wouldn't necessarily always outperform the other methods in that metric.

- The manuscript's branding as minimax, to my mind, necessitates further exploration of the following questions.
    ---> First, the natural conceptual/qualitative question that this manuscript entails but elides is thus: how is it the case that a minimax (and therefore theoretically conservative) approach outperforms other methods that may not have such conservatism explicitly built-in?
   ---> Second, the method is to aggregate the scores by taking their max. Instead, if taking a soft-max (according to some surrogate soft-max), would that help "smooth out" the method in desirable ways?
   ---> Also, as a minor qualm about the paper positioning itself in the context of minimax optimization: E.g. when the intro speaks about the proposed method as "based on minimax optimization", that is a bit confusing/could feel misleading to some; while it's certainly the case that the method solves a minimax optimization problem, it does not actually need to use any of the machinery contained in the minimax optimization literature, due to the very simple structure of the problem in this case. Thus, it would be more sensible to e.g. frame the contribution as stemming from "a minimax formulation", or "a minimax intuition".

---

> ### Author Response · Authors · 2025-09-19
> **Response part 1**
>
> We thank reviewer kgLm for their valuable feedback.
>
> We first highlight a few major changes in our latest revision.
>
> - We added experiments that use synthetic data and do not use posterior sampling.  These experiments also investigate the effect of using too few training and tuning samples.
>
> - In all experiments, we added a variation of our minimax method that uses the QN nonconformity score from Sampson & Chan (2024).  This enables a direct comparison to Sampson & Chan (2024) and demonstrates the flexibility of our minimax method.
>
> - In all experiments, we evaluated the copula-based method of Sun & Yu (2024).
>
> - We added plots and discussion of single-target coverage whereever applicable.
>
>
> Below we respond to each of your questions and concerns.
>
> ## Weaknesses
>
> *While the proposed approach is well-situated by the authors in the context of conformal prediction based tools, that is not so with respect to the slightly more expansive toolkit of distribution-free UQ. In particular, in the multi-target context, the well-known Learn then Test (LTT) method of Angelopoulos et al pioneered a hypothesis-testing framework that helps obtain guarantees that can simultaneously bound multiple notions of (monotonic or nonmonotonic) risk (not restricted to, but including, miscoverage rates). LTT is also in particular well-established in the empirical terrain, including CV applications. Thus, this appears to be a substantial omission.*
>
> The revision includes a discussion of LTT.  Please see below for further details.
>
> *The empirics, while conducted in a (from what I can tell) well-established and motivated medical imaging setting, lack a certain amount of details. For instance, the Sampson and Chan QN method, that is directly compared to, aims to combine the scores into one (similar to the presented method), but they go through the more convoluted step of renormalizing the scores using quantile regression. Their point is to mitigate the disparity between the different dimensions' scores' magnitudes, which at first glance their formula manages to achieve in some sense. However, the minimax approach indeed achieves (at least somewhat) better performance in presented experiments, which makes one wonder: Is this due to the QN method being unable to handle the differing scale or distribution of the different metrics' values?*
>
> We've included a new variation on our method, which we call QN+Minimax, that uses the same nonconformity score of the Sampson & Chan (2024) paper.
> This facilitates a direct comparison to the "QN" method of Sampson & Chan (2024).
>
> *For this and other typical multi-objective reasons, I would have liked to see more visualizations of the different score dimensions, to help the reader better understand the trade-offs that the tested methods have to navigate.*
>
> We've added visualizations of the single-target coverage wherever applicable.
>
> *In a similar vein, I would have liked to see some well-thought-out synthetic experiments on the proposed method comparing it to prior approaches, to better understand: to what extent is the proposed method better than the others due to some special structure in medical imaging (inverse) tasks, and to what extent would that advantage erode under different relative distributions of the different dimensions' scores? In particular, the method doesn't directly control the interval length, and therefore it wouldn't necessarily always outperform the other methods in that metric.*
>
> We've added synthetic-data experiments that demonstrate the distinctive behaviors of each method.
> Indeed, our proposed minimax methods do not always lead to the tightest prediction intervals.
> But they do seem to provide a good balance between the various single-target coverages and, consequently, avoid the situation where some targets get very tight intervals and others very loose intervals.

---

> ### Author Response · Authors · 2025-09-19
> **Response part 2**
>
> *The manuscript's branding as minimax, to my mind, necessitates further exploration of the following questions. ---> First, the natural conceptual/qualitative question that this manuscript entails but elides is thus: how is it the case that a minimax (and therefore theoretically conservative) approach outperforms other methods that may not have such conservatism explicitly built-in?*
>
> The proposed approach aims to outperform competing approaches only in the sense of minimizing the maximum single-target coverage.
> However, it achieves this aim only in the limit of infinite tuning and calibration samples, as we establish in Theorem 1.
> Still, with finite but sufficient tuning data, our experiments suggest that it does a reasonable job of minimizing maximum single-target coverage.
>
> In any case, as the reviewer is well aware, single-target conformal prediction always involves an implicit tradeoff between coverage and interval length.
> With multiple targets, the situation becomes more complicated because there's now a huge space of solutions that all guarantee joint coverage.
> For example, it's possible to obtain simultaneously good coverage and interval length for one target as long as they are simultaneously degraded for other targets.
> Within this vast space of solutions, we postulate that it's worth considering the minimax solution, which aims for balancing coverage across targets in an effort to avoid unnecessarily large prediction intervals.
>
> *---> Second, the method is to aggregate the scores by taking their max. Instead, if taking a soft-max (according to some surrogate soft-max), would that help "smooth out" the method in desirable ways?*
>
> That is an interest idea, but one that is currently out of scope.
>
> *---> Also, as a minor qualm about the paper positioning itself in the context of minimax optimization: E.g. when the intro speaks about the proposed method as "based on minimax optimization", that is a bit confusing/could feel misleading to some; while it's certainly the case that the method solves a minimax optimization problem, it does not actually need to use any of the machinery contained in the minimax optimization literature, due to the very simple structure of the problem in this case. Thus, it would be more sensible to e.g. frame the contribution as stemming from "a minimax formulation", or "a minimax intuition".*
>
> We agree. In the revision, we instead use either "minimax formulation" or "asymptotically minimax".
>
> ## Requested Changes
>
> *Please introduce a detailed discussion comparing to the Learn-then-Test approach and resulting methods based on multiple hypothesis testing rather than the conformal approach. In particular, I'd like for such a comparison to happen from the (a) methodological/expository point of view (i.e. a careful discussion of how LTT achieves its guarantees vs. the proposed method, and in particular a discussion for the readers about the difference in provided guarantees), and (b) experimental point of view (i.e. does the proposed method achieve less conservative coverage/length than LTT?).*
>
> The Learn-then-Test (LTT) framework has the distinct advantage of handling multiple risks.  But in our formulation, there is only one risk (i.e., empirical joint coverage), in which case the LTT coverage guarantee takes the form
>
> $$  \text{Pr} \big[ \text{Pr} \big\\{ \cap_{k=1}^{K} Z\_{0,k} \in \mathcal{C}\_{\hat{\lambda}(D\_{\mathsf{cal}}),k}(\hat{Z}_{0}) \big| D\_{\mathsf{cal}} \big\\} \geq 1-\alpha \big] \geq 1-\delta $$
>
> where $\alpha, \delta \in (0,1)$ are both user-selected error rates.
> Due to the difference in coverage guarantees, the intervals produced by LTT are not directly comparable to intervals produced by any of the methods under consideration in our paper.
> Thus, there is no fair way to incorporate LTT into our experiments.
> We've included a discussion of LTT in Section 2.3.
>
> *Empirically, I would like to request extra visualizations that would (a) better showcase the tradeoffs induced by the multiple objectives, to better understand how the different methods that are compared deal with them; and (b) compare the methods on another task, preferably a well-designed synthetic one, in particular to better delineate the limitations of the proposed method due to its conservatism.*
>
> We've included visualizations of single-target coverage, wherever applicable.
> We've added synthetic-data experiments that demonstrate the distinctive behaviors of each method.
> Furthermore, the synthetic-data experiments include investigations into how the numbers of training and tuning samples affect single-target coverage.

---

> ### Author Response · Authors · 2025-09-19
> **Response part 3**
>
> *A more focused discussion of the minimax nature of the method as per the last of the Weaknesses; this would help the readers navigate (a) the interesting contrast between a robust nature of the proposed method vs. its more efficient performance than the alternatives; and (b) understand future work avenues, such as those stemming from replacing max -> softmax.*
>
> We believe the new plots and discussions around the single-target coverage help to clarify the objectives of our minimax formulation and demonstrate the tradeoffs that each method makes in interval length and coverage.
> As discussed earlier, our method does not aim to provide tighter prediction intervals or higher coverage for all targets, but rather to minimizes the maximum single-target coverage, and it does so only in the large-sample regime.

---

### Review · Reviewer_Fe5k · 2025-08-01

**Summary Of Contributions:**

The authors present a novel way to apply conformal prediction for obtaining multidimensional prediction interval that, on average, cover the true values of multiple predictions concurrently. They develop and apply this technique specifically in the context of downstream tasks on image reconstructions. Specifically, for MRI acceleration reconstructions, they show that their method generally produces smaller intervals than existing methods while still achieving joint coverage.

The heart of their technique is a minimax optimization formulation in which the task with the largest coverage is minimized while still ensuring joint coverage. They develop an algorithm that approximates a solution to this optimization and correct it to produce valid intervals in finite samples using split conformal prediction. They also prove that the algorithm produces the true solution given infinite samples. They then apply this technique for 3 different experiments: multi-metric coverage, multi-task coverage, and multi-round acquisition.

To validate their method, they run each of these experiments in the context of upsampling undersampled MRIs. They train and test a reconstruction network, posterior sampler, and classifier on the fastMRI dataset. They show that they can achieve joint coverage while also minimizing the average set size, improving upon previous techniques.

**Audience:**

Yes

**Broader Impact Concerns:**

As a minor concern, as often comes up with conformal prediction work, I think it is imperative to address the limitations of the guarantees that conformal provides. Though having rigorous uncertainty quantification is extremely positive for scientists, it is also critical for them to understand the nature of the prediction intervals (i.e., marginal vs conditional coverage) and how it applies to their specific context. To that end, it may be worth including a small discussion in the broader impacts statements that gives an example of how exactly an MRI scientists should interpret and utilize the results given in this manuscript.

**Claims And Evidence:**

Yes

**Requested Changes:**

Major concerns (needed for acceptance)
- I believe the authors should report per-target marginal coverage. That is, report the coverage stratified by each task/metric separately.
    - Though the minimax formulation helps prevent coverage imbalance between the targets, it can still be the case that  one target could still achieve a much higher coverage than others, especially because there is only a single conformal threshold over all targets. The paper does not report marginal coverage per target, so it's difficult to assess whether this occurs in practice. Since metrics/tasks can vary significantly in difficulty, this could be a very real possibility. Indeed, the classification performance in the MRI task example varies widely due to the class imbalance issue.
- I’m also a bit concerned with the reliance on posterior sampling. Many of the methods presented here require access to posterior samples, which may or may not be accurate and could also be difficult to obtain. This merits a short discussion about the implications of bad posterior samples and alternative strategies if such a sampler is not attainable (e.g., directly doing quantile regression with the task networks).

Suggestions (not needed for acceptance but would strengthen)
 - The notation can be a bit verbose at time. I think certain things like defining h_k may be unnecessary since the predictions /hat{z} are already defined and shown as the output of task /mu_k or metric m.
 - All the subsections of section 4 contain very similar language (like the last sentence of each subsection). This can be tightened and condensed.
 - Details about the performance of the metric estimation task would also be valuable, similar to what was done for the task case. That way trends about the coverage performance/set sizes can be correlated with prediction accuracy.
 - I think a more challenging experiment would be valuable to display. For example, covering a metric and task simultaneously is potentially challenging because of the difference in scales and the complex dependency between them but should be something this method could handle. It would also be of great value to understand to what degree making good quality reconstructions affects classification accuracy. Especially if you also trained a network that directly classified from reconstruction images instead of clean images.

Minor Clarifications
- In section 4.3, what are the meainging of Z_1, Z_2, Z_3,…(which you assume are exchangeable) My understanding is that the goal is to predict a single Z_0 through the procedure? Are these the calibration/tuning data points?
- Why use a classifier that is trained on clean images, shouldn’t you use one trained on the images from the reconstruction methods?
- Why is your method worse for the meniscus tear classification?
    - Is this at all correlated with the class imbalance (meniscus tears are overrepresented)

**Strengths And Weaknesses:**

Strengths:
- The paper offers a theoretically-justified and experimentally effective strategy for obtaining multidimensional intervals with joint coverage
- They avoid making significant assumptions like task independence and provide a general framework for combining together a wide variety of tasks
-Their minimax formulation is well motivated in preventing severe coverage imbalance while simultaneously allowing for a simple and tractable algorithm
-Their proof in demonstrating convergence in infinite samples also helps justify the algorithm as principled.

Weaknesses/Clarifications (these will be further specified in the changes section):
- In general, the evaluation of the method is fairly barebones and could be improved through additional experiments
- The paper could use more comprehensive evaluation, especially in stratifying metrics across relevant groups (e.g., tasks, interval sizes)
- The paper could include additional discussion on things like task balance, the implications of using a single parameter to control multiple coverages, and the consequence induced by using sample statistics in multiple areas (sample CDF, posterior sampler, etc). A discussion on tuning set size and it's effect is also warranted.
-Another example is deploying more interesting settings like simultaneously covering a metric and downstream task.

---

> ### Author Response · Authors · 2025-09-19
> **Response part 1**
>
> We thank reviewer Fe5k for their valuable feedback.
> We appreciate that the reviewer acknowledges that our method is principled and theoretically justified.
>
> We first highlight a few major changes in our latest revision.
>
> - We added experiments that use synthetic data and do not use posterior sampling. These experiments also investigate the effect of using too few training and tuning samples.
>
> - In all experiments, we added a variation of our minimax method that uses the QN nonconformity score from Sampson & Chan (2024).  This enables a direct comparison to Sampson & Chan (2024) and demonstrates the flexibility of our minimax method.
>
> - In all experiments, we evaluated the copula-based method of Sun & Yu (2024).
>
> - We added plots and discussion of single-target coverage wherever applicable.
>
> - Throughout the paper, we added many clarifications to the text
>
> Below we respond to each of your questions and concerns.
>
> ## Weaknesses
>
> *In general, the evaluation of the method is fairly barebones and could be improved through additional experiments.*
>
> Many additional experiments are reported in the revision.
> For example, there is a new experiment that uses synthetic data, and two new methods have been evaluated in every experiment: the copula-based method of Sun and Yu (2024) and a variation of our proposed minimax method that uses a different nonconformity score.
>
> *The paper could use more comprehensive evaluation, especially in stratifying metrics across relevant groups (e.g., tasks, interval sizes).*
>
> We've added evaluations of single-task coverage wherever applicable.
>
> *The paper could include additional discussion on things like task balance, the implications of using a single parameter to control multiple coverages, and the consequence induced by using sample statistics in multiple areas (sample CDF, posterior sampler, etc). A discussion on tuning set size and it’s effect is also warranted.*
>
> In the revision, we've added discussions of task balance to Sections 1, 2, and 3.2, emphasizing that our minimax method aims to balance the single-target coverage across classes.
> The experiments in Section 5 give ample evidence that it succeeds in this aim.
> Near the end of Section 3.1, we've broadened the discussion of using a single parameter, clarifying that we can do so without loss of minimax optimality.
> As for analyzing the effect of using an empirical CDF and empirical quantiles, this is the goal of Section 3.3, which establishes that our proposed scheme is indeed minimax in the large-data regime.
> As for the effect of using an approximate posterior sampler, we've added a discussion just after Eq.(30) clarifying that, as the sampler gets worse, the prediction intervals loosen but the coverage guarantees still hold.
> Finally, the effect of tuning-set size is now investigated in our synthetic data experiment.
>
> ## Requested Changes
>
> ### Major concerns (needed for acceptance)
>
> *I believe the authors should report per-target marginal coverage. That is, report the coverage stratified by each task/metric separately. Though the minimax formulation helps prevent coverage imbalance between the targets, it can still be the case that one target could still achieve a much higher coverage than others, especially because there is only a single conformal threshold over all targets.*
>
> Yes, we agree that single-target coverage provides important insights.
> We now report the expected single-target coverage "ESC$\_k$" in all experiments (see Figs. 1,2,3,5,6).  These plots give evidence that the minimax formulation does indeed produce low maximum single-target coverage when given sufficient tuning samples.
>
> As for using a single conformal threshold, we can do this without any compromise of the minimax objective, as we now discuss after (22).
>
> *I'm also a bit concerned with the reliance on posterior sampling. Many of the methods presented here require access to posterior samples, which may or may not be accurate and could also be difficult to obtain. This merits a short discussion about the implications of bad posterior samples.*
>
> To be clear, the proposed minimax methodology from Section 3 does not rely on posterior sampling.
> For example, the synthetic-data experiments in Section 5.1 do not involve posterior sampling but rather use quantile regressors.
>
> That said, the imaging applications in Section 4 do use approximate posterior sampling.
> Importantly, the corresponding coverage guarantees hold for *any* approximate posterior sampler, no matter how bad, although the prediction intervals are expected to get looser as the sampler gets less accurate.
> A discussion of posterior sampling has been added just after Eq.(30).

---

> ### Author Response · Authors · 2025-09-19
> **Response part 2**
>
> ### Suggestions (not needed for acceptance but would strengthen)
>
> *The notation can be a bit verbose at time. I think certain things like defining $h_k$ may be unnecessary*
>
> Thanks for this recommendation.
> We've revised Section 4 in accordance and removed mention of $h(\cdot)$ after page 2.
> If you have any other specific recommendations, please let us know.
>
> *All the subsections of section 4 contain very similar language (like the last sentence of each subsection)*
>
> Thanks, we've revised Section 4 based on this recommendation.
>
> *Details about the performance of the metric estimation task would also be valuable, similar to what was done for the task case.*
>
> We've expanded Section 5.2.1 (on performance of FRIQ metric estimation) to include more details.
> For example, we report that the average FRIQ performance of the reconstruction methods (E2E-VarNet and CNF) match those reported in Wen et al. (2025).
>
> *I think a more challenging experiment would be valuable to display. For example, covering a metric and task simultaneously...*
>
> Yes, that could be an interesting experiment, but please note that the experiment in Section 5.2.4 is actually more challenging, in that it simultaneously involves multiple measurement rounds and multiple pathology labels.
> In any case, the revision contains many more experiments than the original submission, and we hope that this makes it more suitable for acceptance.
>
> ### Minor Clarifications
>
> *In section 4.3, what are the meaning of $Z\_1, Z\_2, Z\_3 ,...$ (which you assume are exchangeable)? Are these the calibration/tuning data points?*
>
> Yes, $Z\_1, Z\_2, Z\_3 ,...$ are calibration/tuning samples and $Z\_0$ is the test sample.
> The set $\{Z\_0, Z\_1, Z\_2,...\}$ must be statistically exchangeable for the coverage guarantee to hold.
>
> *Why use a classifier that is trained on clean images, shouldn’t you use one trained on the images from the reconstruction methods?*
>
> There are a number of reasons why one would not want to do this.
> First, the reconstructed images have errors/artifacts that would degrade the classifier and possibly even prevent successful classification.
> Second, the classifier would depend on the specific combination of measurement environment and reconstruction method, and would behave unpredictably for other combinations.
> Third, by training a clean-image classifier, we can use it to form a reconstruction-quality metric that can be used with arbitrary combinations of measurement and reconstruction schemes.
> For example, in the multi-round procedure, we gradually collect measurements only until the uncertainty in the (clean-image) classifier output is below a user-selected level.
> As an alternative, one could imagine using fixed measurements but trying a series of image-reconstruction methods, starting with a computationally cheap one and progressing to more expensive ones, stopping as soon as the uncertainty in the (clean-image) classifier output is below a user-selected level.
>
>
> *Why is your method worse for the meniscus tear classification? Is this at all correlated with the class imbalance (meniscus tears are overrepresented)*
>
> In the multi-task experiment, it is difficult to say why our mean interval lengths (MILs) are worse than the competition for meniscus-tears but better for other pathologies, since each method constructs the intervals and provides coverage in a different way.
> Class imbalance likely accounts for the higher classification performance on meniscus tears, which may correlate with smaller prediction intervals.
>
>
> ## Broader Impact Concerns
> *As a minor concern, as often comes up with conformal prediction work, I think it is imperative to address the limitations of the guarantees that conformal provides. To that end, it may be worth including a small discussion in the broader impacts statements that gives an example of how exactly an MRI scientists should interpret and utilize the results given in this manuscript.*
>
> Thank you for this recommendation.
> We've extended the Broader Impact Statement to provide an example of how MRI scientists should interpret our results.

---

### Review · Reviewer_VTCb · 2025-09-09

**Summary Of Contributions:**

The submission studies conformal prediction (CP) in imaging inverse problems. In particular, it focuses on joint marginal coverage of multiple quality or task-related metrics (e.g., PSNR, SSIM, post-softmax outputs of a classifier). The submission proposes an optimization formulation of multi-target CP, based on the intuition that coverage is monotonically non-decreasing with the size of the prediction sets. A finite-sample procedure is designed, which retrieves the solution of the optimization problem asymptotically.

Experiments are performed on fast MRI data, for both reconstruction and downstream pathology classification. Results show that the proposed method provides coverage with generally tighter uncertainty intervals.

**Audience:**

Yes

**Broader Impact Concerns:**

No concerns

**Claims And Evidence:**

Yes

**Requested Changes:**

**Connections with Copula-based methods**
Could the authors expand on the relation of the proposed method with existing copula-based CP alternatives? The optimization formulation in Eq. (20) is very similar to Eq. (8) in Sun et al., "Copula-based CP for Multi-step Time Series Forecasting" (2022).

Furthermore, it is stated in Sec. 2.3 that "there is no guarantee of satisfying a coverage guarantee like (9) with a finite calibration set". This claim is true of Messoudi et al. (2021), who do not provide a finite-sample theorem. However, Sun et al. do show how to calibrate the copula estimation with finite samples (see Theorem 4.1). In particular, Conformal Predictive Distributions do allow to obtain finite-sample guarantees on estimates of the marginal empirical CDFs (see "Nonparametric predictive distributions based on conformal prediction", Vovk et al. (2017)).

It is important to compare the performance of the proposed approach with existing copula-based methods.

**Motivation of proposed method**
If the core contribution of the submission is the minimax formulation, this should be emphasized and motivated better. Currently, the method is motivated by the observation that coverage is monotonically non-decreasing with the size of the prediction sets, which feels somewhat reductive. I think there are deeper, important considerations to discuss. For example, what happens with the current formulation if each "target" represents a different demographic? What if different FRIQ metrics have different importance for the downstream task at hand?

**Finite-sample guarantees**
As mentioned above, I think there might exist finite-sample algorithms that would provide coverage. In its current formulation, Theorem 1 falls short of delivering on the promise of the method.

**Multi-round measurement acquisition**
Could the authors expand on the observation that "we really desire that the multi-round coverage is at least $1 - \alpha$"? In imaging, for example, why do we want coverage to hold at each step? Intuitively, we want coverage to hold when we make a decision, i.e. at the end? Also why is multi-round coverage expressed as the sum of the single-round coverages? Again, this feels very close in spirit to Sun et al., where they also have a discrete time horizon, and the goal is to provide uniform coverage over time steps.

**A few relevant works on multivariate conformal prediction**
It might be useful to include and discuss these citations to better place the contributions of the current submission within the broader CP research subfield:

[1] "Multivariate Conformal Prediction using Optimal Transport", Klein et al. (2025)

[2] "Optimal Transport-based Conformal Prediction", Thurin et al. (2025)

[3] "Multivariate Conformal Prediction via Conformalized Gaussian Scoring", Braun et al. (2025)

[4] "Minimum Volume Conformal Sets for Multivariate Regression", Braun et al. (2025)

[5] "Length Optimization in Conformal Prediction", Kiyani et al. (2024)

[6] "Fast Nonlinear Vector Quantile Regression", Rosenberg et al. (2023)

---

**Experiments**

In models, both an "image recovery model $f$" and a posterior sampling method $g$ are mentioned. Could the authors expand on how each is used for each calibration method? Do all method have access to the same predictions, or are there differences that could confound the results?

In Sec. 5.2, it is stated that "the goal is to ascertain the presence/absence of each of $K=5$ different pathologies", which would imply hard binary labels. Later, it is mentioned "the true soft-output $z_{0,k}$". Could the authors expand on where the soft-output ground-truth labels come from?

Could the authors expand on the SC method and why it fails to provide coverage?

In the multi-round experiment, how are the thresholds $\tau = 0.16$ and $\tau = 0.10$ chosen? In the current form, these feel arbitrary, and might introduce bias in the results. Have the authors considered plotting mean interval length as a function of acceleration? Is there any guarantee that the interval length will be decreasing with the number of measurements?

---

**Minor comments**
- "and radiologists receive ...". This sentence feels a little out of context, maybe introduce with "For example"?
- "only a single target", "a scalar target" would be more precise
- "with certain statistical guarantees", it is unclear whether this sentence refers to "uncertainty intervals" or "black-box predictor".
- "only the conformalized quantile regression (CQR) method of Romano et al.". I am not sure I follow how Eq. (6) is the same as CQR. In CQR, one trains a predictor with the pinball loss, and subsequently calibrate such predictor. Here, calibrated quantiles are taken from empirical samples, without training a predictor. Furthermore, CQR uses a multiplicative parametrization of the prediction sets, but Eq. (7) uses an additive parametrization.
- DISTS is not defined.
- In Fig. 3, the labels of the axes may be misinterpreted by the subcaptions. I would suggest removing the subcaptions, since the information is already presented in the y-axis labels.
- "they are less interpretable, since the uncertainty intervals ... the other target components". This claim is task-dependent. Sometimes, the dependence structure across different metrics is fundamental to obtain interpretable claims. In imaging, for example, principal components uncover important spatial relations.

**Strengths And Weaknesses:**

**Strengths**
- Uncertainty quantification for imaging inverse problems is an important topic
- Multivariate CP is an open problem
- Interval length minimization is central to CP methods

**Weaknesses**
- Presentation of the proposed method could be expanded on
- Comparison with existing copula-based methods should be included given the similarity of the approach
- Lack of finite-sample guarantees of the proposed method

I will expand on these points below and I am looking forward to discussing with the authors!

---

> ### Author Response · Authors · 2025-09-19
> **Response part 1**
>
> We thank reviewer VTCb for their valuable feedback.
>
> We first highlight a few major changes in our latest revision.
>
> - We added experiments that use synthetic data and do not use posterior sampling. These experiments also investigate the effect of using too few training and tuning samples.
>
> - In all experiments, we added a variation of our minimax method that uses the QN nonconformity score from Sampson & Chan (2024).  This enables a direct comparison to the Sampson & Chan (2024) and demonstrates the flexibility of our minimax method.
>
> - In all experiments, we evaluated the copula-based method of Sun & Yu (2024).
>
> - We added plots and discussion of single-target coverage wherever applicable.
>
> Below we respond to each of your questions and concerns.
>
>
> ## Requested Changes
>
> **Connections with Copula-based methods** *Could the authors expand on the relation of the proposed method with existing copula-based CP alternatives? The optimization formulation in Eq. (22) is very similar to Eq. (8) in Sun et al., "Copula-based CP for Multi-step Time Series Forecasting" (2022). Furthermore, it is stated in Sec. 2.3 that "there is no guarantee of satisfying a coverage guarantee like (9) with a finite calibration set". This claim is true of Messoudi et al. (2021), who do not provide a finite-sample theorem. However, Sun et al. do show how to calibrate the copula estimation with finite samples (see Theorem 4.1). In particular, Conformal Predictive Distributions do allow to obtain finite-sample guarantees on estimates of the marginal empirical CDFs (see "Nonparametric predictive distributions based on conformal prediction", Vovk et al. (2017)). It is important to compare the performance of the proposed approach with existing copula-based methods.*
>
> Thanks for pointing out this ICLR'24 paper by Sun and Yu.  Indeed, their Eq. (8) has some similarities to our Eq. (22), but there are at least two major differences, which we clarify here and in the revision.
> First, they minimize the *sum* of the copula inputs, while we minimize the *max* of the copula inputs.  Consequently, their formulation does not provide any incentive for balancing the single-target coverages, while ours does.  In doing so, our approach aims to prevent the situation where one target's coverage is favored over another's.
> Second, their Eq. (8) involves a computationally expensive multi-variable optimization, while ours is easily reduced to the single-variable search in (23).  In practice, they use SGD (see their Appendix B.2), which requires hyperparameter tuning and gives no guarantee of finding a global optimum.
>
> We evaluated Sun and Yu's approach in all of our experiments and added the results to our revision.
> As expected, we see that Sun and Yu's approach gives less balanced single-target coverage than our minimax approach and, consequently, is much more likely to produce overly loose prediction intervals.
>
> **Motivation of proposed method** *If the core contribution of the submission is the minimax formulation, this should be emphasized and motivated better. Currently, the method is motivated by the observation that coverage is monotonically non-decreasing with the size of the prediction sets, which feels somewhat reductive. I think there are deeper, important considerations to discuss. For example, what happens with the current formulation if each "target" represents a different demographic? What if different FRIQ metrics have different importance for the downstream task at hand?*
>
> The core contribution is a minimax-inspired approach to multi-target conformal prediction that strives for balanced single-target coverages.
> Due to the monotonic relationship between coverage and interval size, balancing the single-target coverages helps to prevent unnecessarily large prediction intervals.
> We clarify these points in the revision.
>
> We're not sure how to interpret "demographic" in this context.
> Also, our formulation does not cover the case where some FRIQ metrics are more important than others.
> One could imagine various ways to extend our formulation to handle this case, but doing so is outside the scope of our submission.

---

> ### Author Response · Authors · 2025-09-19
> **Response part 2**
>
> **Finite-sample guarantees** *As mentioned above, I think there might exist finite-sample algorithms that would provide coverage. In its current formulation, Theorem 1 falls short of delivering on the promise of the method.*
>
> There must be some confusion, since our method does indeed provide marginal joint coverage with a finite number of tuning and calibration samples.
> As explained in Section 3.2, our method is an instance of split-conformal prediction (see Section 2.1) and thus enjoys the same benefits.
> Within the family of multi-target split-conformal prediction algorithms, one could say that we propose a nonconformity score that ensures joint-coverage across the components of a multi-dimensional target while also promoting balanced single-target coverage across those components.
>
> The goal of the asymptotic analysis in our Theorem 1 is to connect our finite-sample scheme back to the motivating random-variable formulation in Section 3.1, since it is not clear whether the use of empirical versions of the CDF and quantile functions compromises the minimax formulation from Section 3.1.
>
> **Multi-round measurement acquisition** *Could the authors expand on the observation that "we really desire that the multi-round coverage is at least"? In imaging, for example, why do we want coverage to hold at each step? Intuitively, we want coverage to hold when we make a decision, i.e. at the end? Also why is multi-round coverage expressed as the sum of the single-round coverages? Again, this feels very close in spirit to Sun et al., where they also have a discrete time horizon, and the goal is to provide uniform coverage over time steps.*
>
> Indeed, we want the coverage to hold "at the end", i.e., when a measurement is accepted, and that is exactly what Eq. (31) describes.
> That equation decomposes coverage-at-the-accepted-round into a sum of terms that each handle a different possibility for the accepted round.
> In other words,
> $$ \text{Pr}\\{\text{coverage at accepted round}\\} = \sum\_{b=1}^B \text{Pr}\\{\text{coverage at accepted round \\& accepted round} = b\\}, $$
> which follows from basic probability.
> Note that,
> for a measurement to be accepted at round $2,3,...B-1$, it must have not been accepted at any previous round *and* must be good enough for acceptance at the current round;
> for a measurement to be accepted at round $1$, it must be good enough at that round;
> and for a measurement to be accepted at the final round $B$, it must not have been accepted at any previous round.
> No formulations of this sort are considered in Sun and Yu (2024).
>
> **A few relevant works on multivariate conformal prediction** *It might be useful to include and discuss these citations to better place the contributions of the current submission within the broader CP research subfield:*
>
> *[1] "Multivariate Conformal Prediction using Optimal Transport", Klein et al. (2025).*
>
> The goal is similar to [2] below, but this was rejected by ICML'25 while [2] was accepted, and so we will cite [2].
>
> *[2] "Optimal Transport-based Conformal Prediction", Thurin et al. (ICML'25).*
>
> Yes, we will discuss and cite this paper.
>
> *[3] "Multivariate Conformal Prediction via Conformalized Gaussian Scoring", Braun et al. (arxiv 7/28/25).*
>
> This appeared on arXiv on 28 July 2025, which was after our paper was submitted.
>
> *[4] "Minimum Volume Conformal Sets for Multivariate Regression", Braun et al. (arxiv 3/24/25).*
>
> Yes, we will discuss and cite this paper.
>
> *[5] "Length Optimization in Conformal Prediction", Kiyani et al. (NeurIPS'24).*
>
> This paper seems to focus on conditional coverage and does not explicitly consider multi-target prediction.
>
> *[6] "Fast Nonlinear Vector Quantile Regression", Rosenberg et al. (ICLR'23).*
>
> Yes, we will discuss and cite this paper.

---

> ### Author Response · Authors · 2025-09-19
> **Response part 3**
>
> ### Experiments
> *In models, both an "image recovery model" and a posterior sampling method are mentioned. Could the authors expand on how each is used for each calibration method? Do all method have access to the same predictions, or are there differences that could confound the results?*
>
> Each calibration method is given the same training+calibration samples $\\{(\hat{z}\_{i,k},z\_{i,k})\\}\_{i>0}$ and is evaluated on the same test samples $(\hat{z}\_{0,k},z\_{0,k})$.
> Thus, the image recovery model and posterior sampler are used identically across methods.
> Because the CPTS and minimax methods use tuning samples while the IA and QN methods don't, we allow the IA and QN methods to use the tuning samples as additional calibration samples, for fairness.
>
> *In Sec. 5.2, it is stated that "the goal is to ascertain the presence/absence of each of different pathologies", which would imply hard binary labels. Later, it is mentioned "the true soft-output ". Could the authors expand on where the soft-output ground-truth labels come from?*
>
> The classifier was trained using hard binary labels that indicate the presence or absence of a pathology.
> But, like most modern classifiers, it returns "soft outputs" in the interval [0,1], where higher values indicate more confidence in the presence of the pathology.
> When referring to the "true" soft-output, we mean $z\_0=\mu(x\_0)$, which is the output of the soft-classifier when given the true image $x_0$.
> (See the discussion before Eq. (5).)
> Our goal is to construct a prediction interval that contains $z_0$ with probability $\geq 1-\alpha$.
> In any case, our use of soft outputs follows the problem formulation from Wen et al. (2024), which is detailed in Section 2.2.
> Additional classifier details are provided in Appendix B.
>
> *Could the authors expand on the SC method and why it fails to provide coverage?*
>
> The SC method is the one proposed in Wen et al. (2024), which separately calibrates a unique conformal predictor for each round.
> Thus, if one applied the $b$th predictor to those samples accepted at the $b$th round, it would provide coverage.
> But, as the reviewer pointed out earlier, we instead want that coverage holds at the *accepted* round for each given sample, and the accepted round is apriori unknown.
>
> *In the multi-round experiment, how are the thresholds and chosen? In the current form, these feel arbitrary, and might introduce bias in the results. Have the authors considered plotting mean interval length as a function of acceleration? Is there any guarantee that the interval length will be decreasing with the number of measurements?*
>
> The thresholds were chosen to match the choices in Wen et al. (2024) and Wen et al. (2025).
> Although the reasons for these choices are explained in the original papers (and in the footnote we've added on page 16), one might still argue that they are somewhat arbitrary.
> Truly meaningful thresholds would require expert guidance and many clinical trials, which is outside the scope of this work.
> Regarding the mean interval length, it was plotted versus acceleration in Wen et al. (2024) for the single-target case and it does indeed decrease with the number of measurements.
> We have not proven that the mean interval length will decrease with the number of measurements, but that is an interesting idea for future work.
>
> ### Minor Comments
> Thank you for your minor comments.  We've integrated most of them in the revision.
>
> *I am not sure I follow how Eq. (6) is the same as CQR.*
>
> Eq.(6) is used to compute the nonconformity score of CQR.  If you compare our Eqs.(6)-(7) to Eq.(9) in Romano et al. (2019), you'll see that they are identical.

---

> > ### Comment · Reviewer_VTCb · 2025-09-26
> > **Thank you for your response!**
> >
> > I sincerely thank the authors for their careful consideration of all reviewers' comments and for including additional experiments that better place the contributions of the submission within the broader uncertainty quantification literature.
> >
> > ## Connections with existing copula based methods
> >
> > Thank you for including the work of Sun et al. and discussing differences, both conceptual and empirical in experiments.
> >
> > ## Motivation of proposed method
> >
> > Thank you for expanding on the message of the proposed method! I agree that minimizing the max may avoid "cheating" by overcovering certain metrics. However, minimizing the maximum coverage does not prevent "giving up" by failing to cover other metrics. I guess this does not happen in the particular experiments studied in the submission because all metrics have the same weight?
> >
> > ## Finite-sample guarantees
> >
> > Thank you for your clarification. Could the authors include a brief lemma that precisely states the guarantees provided by the procedure described in Sec. 3.2? I understand these might follow naturally from standard properties of split conformal prediction. Including a clear statement, however, would solidify the contributions of the paper. A clear algorithm vignette could also be beneficial to crystallize the procedure for readers.
> >
> > ## Multi-round
> >
> > Thank you for your clarification! If I understand correctly, what Eq. (31) - (34) say is:
> >
> > Denote $E_b$ the event "coverage at round $b$", and let $\beta$ be the random stopping round (i.e., the acceptance round), then
> >
> > $P_{multi} = P(E_{\beta}) = P( \bigcup_{b=1}^B \\{A_b \cap \beta = b\\}) = P(\exists b: E_b \cap \beta = b) \geq P(\forall b: E_b) = P(\bigcap_{b=1}^B E_b).$
> >
> > That is, the probability that coverage is satisfied at the acceptance round is greater than the probability that coverage is guaranteed at all rounds.
> >
> > If this is the case, could the authors expand on how this guarantee is different from Theorem 4.1 in Sun et al.? There, they also construct $d$-dimensional regions that provide coverage uniformly over a finite time horizon, although not with the intent of bounding an event at the stopping time of a random process.
> >
> > ## Experiments
> >
> > Thank you for your clarifications. I kindly ask these aspects be made clear in the revised version of the paper in order to avoid misunderstandings.

---

> > > ### Author Response · Authors · 2025-09-27
> > > **Latest response**
> > >
> > > We thank reviewer VTCb for the continued discussion and valuable feedback.
> > >
> > > We first highlight the changes in our latest revision.
> > >
> > > - A lemma detailing our finite-data guarantee has been added.
> > >
> > > - An algorithm detailing our proposed procedure has been added.
> > >
> > > - Clarifications on the experiments have been added.
> > >
> > > The changes in this latest revision are marked in brown (and the previous ones in blue).
> > >
> > > Below we respond to each of your questions and concerns.
> > >
> > >
> > > **Motivation of proposed method** *Thank you for expanding on the message of the proposed method! I agree that minimizing the max may avoid "cheating" by overcovering certain metrics. However, minimizing the maximum coverage does not prevent "giving up" by failing to cover other metrics. I guess this does not happen in the particular experiments studied in the submission because all metrics have the same weight?*
> > >
> > > Recall that we aim to minimize the maximum coverage subject to a joint-coverage constraint.
> > > The joint-coverage constraint helps to prevent a failure of single-target coverage on any one component.
> > > In this sense, we do not believe that our experimental results were coincidental.
> > >
> > > **Finite-sample guarantees**
> > > *Thank you for your clarification. Could the authors include a brief lemma that precisely states the guarantees provided by the procedure described in Sec. 3.2? I understand these might follow naturally from standard properties of split conformal prediction. Including a clear statement, however, would solidify the contributions of the paper. A clear algorithm vignette could also be beneficial to crystallize the procedure for readers.*
> > >
> > > Thanks for these suggestions.
> > > In the latest revision, we included a lemma that details the finite-sample guarantees.
> > > We also included a statement clarifying that the finite-sample guarantee in Sun and Yu (2024) requires exchangeability of the tuning samples (with the test and calibration samples) whereas ours does not.
> > > And we included an algorithm vignette.
> > >
> > > **Multi-round**
> > > *Thank you for your clarification! If I understand correctly, what Eq. (31) - (34) say is: Denote  the event "coverage at round ", and let  be the random stopping round (i.e., the acceptance round), then
> > > $$
> > > P\_{multi} = P(E\_{\beta}) = P( \bigcup\_{b=1}^B \{A\_b \cap \beta = b\}) = P(\exists b: E\_b \cap \beta = b) \geq P(\forall b: E\_b) = P(\bigcap\_{b=1}^B E\_b).
> > > $$
> > > That is, the probability that coverage is satisfied at the acceptance round is greater than the probability that coverage is guaranteed at all rounds.
> > > If this is the case, could the authors expand on how this guarantee is different from Theorem 4.1 in Sun et al.? There, they also construct $d$-dimensional regions that provide coverage uniformly over a finite time horizon, although not with the intent of bounding an event at the stopping time of a random process.*
> > >
> > > Theorem 4.1 of Sun and Yu (2024) is a generic joint-coverage guarantee that can be applied to any multiple regression task, just like our guarantee and that in Sampson and Chan (2024).
> > > Sun and Yu focused on the application of multi-step time-series forecasting, while we focused on multi-metric, multi-task, and multi-round accelerated MRI applications.
> > > These applications, while all instances of multiple regression, are distinct at the application level.
> > >
> > > That said, if one wants to be picky, Theorem 4.1 in Sun and Yu (2024) appears to require that the tuning samples are exchangeable with the test and calibration samples, which is not required for our coverage guarantee nor that in Sampson and Chan (2024).
> > > And if one wants to be really picky, Theorem 4.1 in Sun and Yu (2024) was proven only under i.i.d test/calibration/tuning samples, which is even more restrictive than exchangeable test/calibration/tuning samples.
> > >
> > > **Experiments**
> > > *Thank you for your clarifications. I kindly ask these aspects be made clear in the revised version of the paper in order to avoid misunderstandings.*
> > >
> > > Thanks for this suggestion.
> > > We've carefully gone through the Experiments section of our rebuttal and updated our revision to ensure that all clarifications in the rebuttal are clearly stated in the revision.

---

> > > > ### Comment · Reviewer_VTCb · 2025-10-01
> > > > **Thanks for your response!**
> > > >
> > > > I sincerely thank the authors for engaging in discussion. All my comments and questions have been addressed.

---

### Decision · Action_Editor_Arx4 · 2025-11-06

**Recommendation:** Accept with minor revision

**Additional Comments:**

In the discussion with reviewers, the connections to [1] and [2] were brought up. The authors indicated that:
>In [1].. The goal is similar to [2] below, but this was rejected by ICML'25 while [2] was accepted, and so we will cite [2].

I find this justification to be inappropriate, given that both works are concurrent. Unless the authors have strong reasons to believe that the work in [1] is factually incorrect, and can share those insights with the AE, I request the authors cite both works in the spirit of attribution of concurrent scientific ideas.

[1] "Multivariate Conformal Prediction using Optimal Transport", Klein et al. (2025).

[2] "Optimal Transport-based Conformal Prediction", Thurin et al. (ICML'25).

**Audience:**

Yes

**Audience Explanation:**

Uncertainty quantification in inverse problems is a relevant problem in modern machine learning, and will be of interest to a large portion of TMLR's audience.

**Claims And Evidence:**

Yes

**Claims Explanation:**

This paper proposes a minimax approach for uncertainty quantification through conformal prediction in the context of multi-target inverse problem settings. The approach seeks to provide tight prediction intervals that provably control uncertainty for a variety of metrics, ensuring marginal coverage. The exposition, proofs, and experiments all support these claims.

All reviewers are supportive of this paper: the minimax formulation is conceptually simple and novel, and delivers nicely on the proposed objective. The empirical validation is also comprehensive, both on synthetic and real data. The discussion with the reviewers has further improved the paper, leading to new discussions and comparisons with other related methods, as well as clarifications in the narrative and notation.

I recommend acceptance of this paper, provided they address a minor comment below.

---

> ### Author Response · Authors · 2025-11-10
>
> Thank you for your positive decision.  We have added the citation to "Multivariate Conformal Prediction using Optimal Transport", Klein et al. (2025), in the submitted camera-ready version.